# Aggregation-induced emission luminogens for image-guided surgery in non-human primates

Danni Zhong[1,2,12], Weiyu Chen[3,4,12], Zhiming Xia[5,12], Rong Hu[6,12], Yuchen Qi[2,12], Bo Zhou[2], Wanlin Li[2], Jian He[2], Zhiming Wang[6], Zujin Zhao [6], Dan Ding [7], Mei Tian [8], Ben Zhong Tang [6,9 ✉] & Min Zhou [1,2,10,11 ✉]

During the past two decades, aggregation-induced emission luminogens (AIEgens) have been intensively exploited for biological and biomedical applications. Although a series of investigations have been performed in non-primate animal models, there is few pilot studies in non-human primate animal models, strongly hindering the clinical translation of AIE luminogens (AIEgens). Herein, we present a systemic and multifaceted demonstration of an optical imaging-guided surgical operation via AIEgens from small animals (e.g., mice and rabbits) to rhesus macaque, the typical non-human primate animal model. Specifically, the folic conjugated-AIE luminogen (folic-AIEgen) generates strong and stable fluorescence for the detection and surgical excision of sentinel lymph nodes (SLNs). Moreover, with the superior tumor/normal tissue ratio and rapid tumor accumulation, folic-AIEgen successfully images and guides the precise resection of invisible cancerous metastases. Taken together, the presented strategies of folic-AIEgen based fluorescence intraoperative imaging and visualization-guided surgery show potential for clinical applications.

[1] Eye Center, the Second Affiliated Hospital, Zhejiang University School of Medicine, Hangzhou 310009, China. [2] Institute of Translational Medicine, Zhejiang University, Hangzhou 310009, China. [3] The Fourth Affiliated Hospital, Zhejiang University School of Medicine, Yiwu 320000, China. [4] Molecular Imaging Program at Stanford, Department of Radiology, Stanford University, Stanford 94305, USA. [5] Department of Nuclear Medicine, Shandong Provincial Hospital Affiliated to Shandong First Medical University, Jinan 250021, China. [6] NSFC Center for Luminescence from Molecular Aggregates, SCUT-HKUST Joint Research Institute, State Key Laboratory of Luminescent Materials and Devices, South China University of Technology, Guangzhou 510640, China. [7] Key Laboratory of Bioactive Materials, Ministry of Education, and College of Life Sciences, Nankai University, Tianjin 300071, China. [8] Department of Nuclear Medicine, the Second Affiliated Hospital, Zhejiang University School of Medicine, Hangzhou 310009, China. [9] Department of Chemistry, Hong Kong Branch of Chinese National Engineering Research Center for Tissue Restoration and Reconstruction, State Key Laboratory of Neuroscienceand Division of Biomedical Engineering, The Hong Kong University of Science and Technology (HKUST), Clear Water Bay, Kowloon, Hong Kong, China. [10] Cancer Center, Zhejiang University, Hangzhou 310009, China. [11] State Key Laboratory of Modern Optical Instrumentations, Zhejiang University, Hangzhou 310058, China. [12]These authors contributed equally: Danni Zhong, Weiyu Chen, Zhiming Xia, Rong Hu, Yu chen Qi. ✉email: tangbenz@ust.hk; zhoum@zju.edu.cn

Cancer is one of the leading causes of human death all over the world. Among general therapies, the surgical operation has been widely applied in treating cancer by physically removing malignant tumors. A successful resection is highly depended on the rapid and accurate localization of malignant tissues, especially for treatments of tiny foci and sentinel lymph nodes (SLNs)[1]. More importantly, the physical diagnosis during oncologic surgery, such as palpation and visual inspection are strongly related to surgeons' experience that would eventually affect the outcome of surgery. Compared with the other conventional imaging modalities including X-ray, computed tomography (CT), magnetic resonance imaging (MRI), and ultrasound, the immune-positron emission tomography (Immuno-PET) offers a better preoperative diagnosis via tumor-specific imaging[2]. However, the lengthy acquisition time and ionizing radiation risk greatly restrict its intraoperative application[3]. A powerful image guidance would not only greatly promote the outcome of surgery but also increase the surgical efficiency via a precision operation (e.g., the certain SLNs biopsy & resection) instead of comprehensive surgery, such as whole lymph nodes excision.

The procedure of cancer surgery and outcomes could be potentially improved via fluorescence image-guided surgery (IGS)[4,5]. Since now, various fluorescence molecules have been systemically investigated as imaging probes for optically guided surgery, with high sensitivity, bio-compatibility, portability, and cost-effectively[6–8]. Among all, indocyanine green (ICG), as the most common dye could offer a desirable signal-to-background ratio and tissue-penetration depth, which has been widely employed in clinical applications[4,9,10]. Nevertheless, the ICG-based imaging is also subjected to the finite quantum yield[4], short-term retention in tumor[11], and nonspecific interaction with cells[12]. Besides, the special camera and systematic training required for NIR probes usage further restrain the promotion of ICG[13]. Therefore, versatile intraoperative imaging modalities with tumor-specific targeting are in urgent need.

In the comparison of enhanced permeability and retention (EPR) effect that passively facilitates the accumulation of agent in the tumor area, the tumor-targeting efficiency of imaging probe could be greatly enhanced via ligand-functionalization. Folic acid (FA), also known as vitamin B9 could actively bind to the folate receptor (FR) that was overexpressed in various cancer types[11]. Additionally, a growing body of evidence has suggested that the overexpression of FR is a negative prognostic factor for breast[12], ovarian[13], and colorectal cancer[14]. As an excellent therapeutic target, FR provides an effective option for personalized cancer treatment. The modification of folic acid could induce the endocytosis of imaging agents and gradually detach from folate receptor when the pH decreases, eventually leading to an efficiently targeting and accumulation within tumor tissues[15]. Various folic acid conjugations have been applied in targeted imaging and therapy of FR-expressing malignant cells and tissues[16,17].

In 2001, we discovered an uncommon luminogen system, aggregation-induced emission (AIE) system, rather than destructively as in the conventional systems[18]. For the last 20 years, AIE luminogens (AIEgens) research has made great strides in biological imaging and disease theranostics[19,20]. The aggregation of AIEgens could generate a strong fluorescence emission, while the aggregation-caused quenching (ACQ) effect gradually decreases the signal of conventional fluorophore after accumulation in tumor[21]. It is notable that AIEgens not only provide the high contrast of tumor-to-normal tissue (T/N) but also exhibit a desirable bio-compatibility for in-vivo applications. For instance, near-infrared AIEgen (PTZ-BT-TPA) based fluorescence imaging-guided cancer surgery could delineate tiny tumor nodules and significantly improving the cancer surgery outcome[22]. Moreover, AIE-featured TPETP-SS-DEVD-TPS-

cRGD were able to act as promising photosensitizers for image-guided photodynamic therapy (PDT), with bright emission and high ROS generation[23].

However, most AIEgens applications were limited in in-vitro assays and small animals (such as mice and rats), which strongly withholds AIEgens from clinical translation due to differences in interspecies. The nonhuman primate animal model plays an important role in determining the risks and benefits of nanoprobes for biomedical applications and their eventual clinical translation[24,25]. Hence, the preclinical study on species sharing similarity to humans in genomics, anatomy, physiological functions, and genetic characteristics such as nonhuman primate rhesus macaques is highly demanded before a real transformation[26,27].

In this study, we successfully synthesized folic acid-functionalized AIEgen (folic-AIEgen) that could emit a bright yellow fluorescence (~540 nm) under 365 nm UV lamp irradiation (Fig. 1a). As-prepared folic-AIEgen could actively target folate receptor type alpha (FRα) overexpressing SKOV3 and Hela cancer cells and tissues with mild cytotoxicity. Furthermore, folic-AIEgen was able to perform SLNs biopsy from mice to rhesus macaque and image-guided surgery for tiny SKOV3 and Hela tumors after peritoneal dissemination. Our findings indicate that folic-AIEgen could serve as an effective and convenient fluorescent probe for SLNs biopsy and precisely detection of tiny tumors. All these preclinical studies would greatly accelerate the process of AIEgens in clinical translation (Fig. 1b).

## Results

**High biocompatible and FR-targeting abilities of folic-AIEgen.** The selected AIEgen, 4,7-bis[4-(1,2,2-triphenylvinyl) phenyl] benzo-2,1,3-thiadiazole (BTPEBT, Fig. 1c), has been reported to possess the unique photophysical property with bright emission and good photostability[28]. In order to enhance its dispersity and selectivity, the matrices of DSPE-PEG$_{2000}$-Folic and DSPE-PEG$_{2000}$ were employed to encapsulate BTPEBT, affording the folic-AIEgen nano-probe[29]. Transmission electron microscopy (TEM) images revealed that the as-prepared folic-AIEgen nanoparticles were spherical with an average diameter of ~18 nm and had a good mono-dispersity (Supplementary Fig. S1a). The average hydrodynamic diameter of folic-AIEgen was ~20.3 ± 1.9 nm as measured by dynamic light scattering (DLS) (Supplementary Fig. S1b). The UV-photoluminescence (PL) spectra of the resultant folic-AIEgen demonstrated resolved absorption peaks and symmetrical PL peaks (Supplementary Fig. S1c, d). Notably, the emission window of the as-prepared folic-AIEgen was located in visible wavelength (450–700 nm). A mutifunctional image-guided probe with desirable bio-compatibility is highly demanded in addition to the potent luminous ability. Therefore, the potential cytotoxicity of folic-AIEgen was investigated in various cell lines, including HEK293, HL7702, 4T1, and fibroblast cell lines. After a 24 h incubation of folic-AIEgen (up to 100 μg/mL), all cells' viability reached over 90%, clearly indicating the low cytotoxicity of folic-AIEgen (Supplementary Fig. S2).

Folate receptor (FR) shows a strong affinity with folic acid (Kd ~ 0.1 nM), which has been comprehensively studied and identified as a therapeutic strategy[30]. To evaluate the specific binding of folic acid to FR, cellular imaging of folic-AIEgen was performed in both human ovarian cancer cells, SKOV3 (with overexpression of FR) and keratinocyte cancer cells, HaCaT (with low expression of FR). Rare fluorescence could be found in HaCaT cells after 24 h incubation with folic-AIEgen (Supplementary Fig. S3). In contrast, folic-AIEgen was effectively internalized by SKOV3 cells, with clear intensive fluorescent

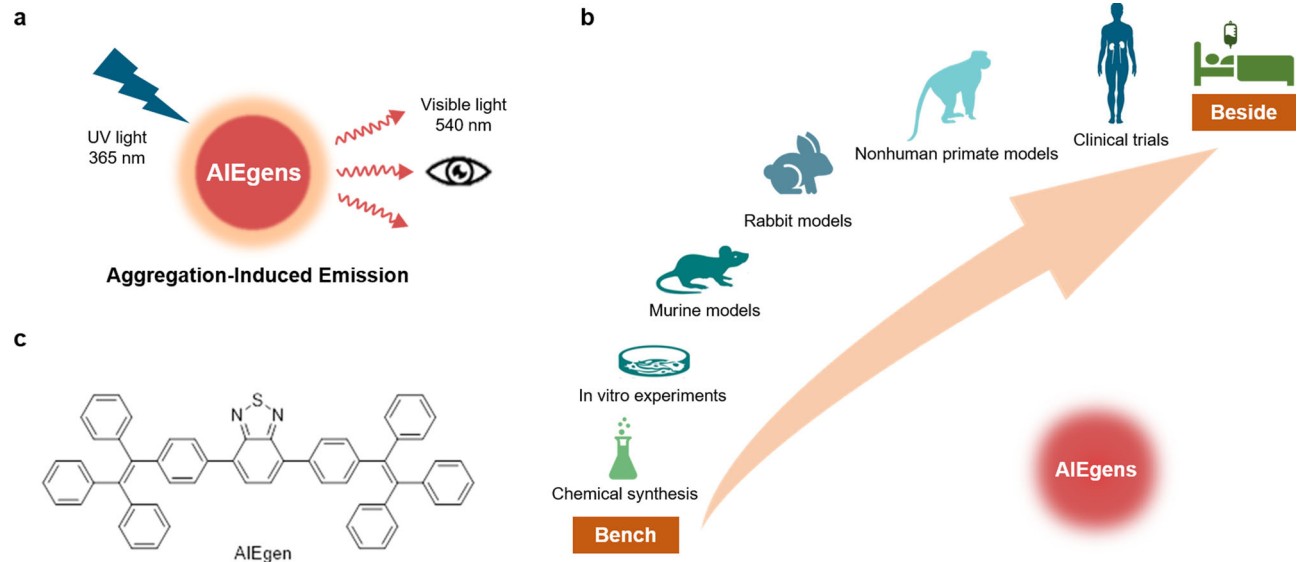

**Fig. 1 From bench to bedside of AIEgens. a** Optical characteristics of AIEgens. **b** Schematic illustration of AIEgens for SLNs biopsy and detection of tiny tumors, from murine, rabbit, to nonhuman primate models, and their potential value for clinical translation. **c** Chemical structure of AIEgen.

signals inside or around the cells. To further confirm that the cellular uptake of folic-AIEgen was dependent on FR-mediated endocytosis, free folic acid was employed as a competitor for blocking the binding site of FR with folic-AIEgen. It is obvious that the intracellular fluorescence was almost abolished with the presence of free FA, suggesting that FR-mediated endocytosis was critical for the internalization of folic-AIEgen in SKOV3 and other tumor cells with high-expressed FR.

**Effective SLN biopsy in nude mice and rabbit breast via folic-AIEgen based image-guided operation.** The feasibility of folic-AIEgen mediated fluorescence visualization under UV-light illumination for real-time SLN detection and resection was first investigated in a nude mice model (Fig. 2a). Folic-AIEgen (25 μL, 40 μg/mL) was subcutaneously (s.c.) injected into the front footpad of nude mice for real-time visualization of lymphatic drainage patterns. Upon injection, folic-AIEgen drains into lymphatic vessels and then into connected popliteal LN. Typically, about 30 s postinjection, the lymphatic vessel, and axillary LN were gradually visible, with a naked-eye visible fluorescence of folic-AIEgen under 365 nm UV irradiation (Fig. 2b, i&ii). The photoluminescence of folic-AIEgen remained stable and strong under UV illumination, allowing the draining LN to be excised precisely and completely under the real-time guidance of bright fluorescence (Fig. 2b, iii). The draining LNs presented significantly enhanced fluorescent signals ($13.73 * 10^8$ [p/s]/[μW/cm²]), which was about 4 times of that in contralateral LNs ($P < 0.001$) (Fig. 2c, d). The rate of nanoparticle diffusion in inter-stibium and lymphatic systems is vital for recognizing SLNs[31]. Therefore, the good dispersity of folic-AIEgen would greatly promote its accumulation in SLNs via lymph system and the bright fluorescence generated by folic-AIEgen could help surgeons efficiently identify SLNs for precision surgery.

Unlike popliteal LNs bunching up in popliteal fossa, the axillary LNs of rabbit divide into many clusters and are embedded in the fat, which greatly pulls up the difficulty of localization surgical resection. To further demonstrate the reliability of folic-AIEgen based fluorescence visualization for surgical navigation, real-time SLN detection and resection were carried out in a rabbit model (Fig. 2e). Before injection, a small incision of skin was made to expose the area of the mammary lymphatic system. With a UV lamp's assistance (365 nm), strong fluorescence signals of folic-AIEgen were observed immediately along the lymph vessel

after subcutaneous administration around the papilla (Fig. 2f, i–iii, Supplementary Video 1). After 3 mins postinjection, the lymph vessel was further brightened by the strong fluorescence of folic-AIEgen, while the connected LNs were completely unrecognizable hidden in adipose tissues. Subsequently, the SLN could be accurately identified, dissected out, and excised by tracing the folic-AIEgen fluorescence under UV illumination (Fig. 2f, iv&v). As shown in Fig. 2f(vi) and Supplementary Video 1, SLN biopsy could be performed accurately in real-time, showing significantly strong signals in SLN compared with contralateral LN (Fig. 2g, h). Furthermore, folic-AIEgen was distributed homogeneously in SLN, showing its desirable mobility within the lymphatic system. More importantly, the widely distributed folic-AIEgen did not cause any morphologic changes to SLNs, indicating the high biocompatibility of folic-AIEgen (Fig. 2h).

Afterward, histological analysis was performed to determine whether folic-AIEgen itself or its degradation products would cause tissue damage, inflammation, or lesions in vivo. As shown in Supplementary Figs. S4, 5, no apparent histopathological abnormalities or lesions were observed in the brain, heart, liver, spleen, lung, and kidney from the mice and rabbits treated with folic-AIEgen, preserving the same structures as those of control groups.

Subsequently, we evaluated the safety of surgical procedures and the dose of UV irradiation used during the surgery. The mice in surgery and surgery + UV illumination groups (mice were exposed to 15 m of UV light) behaved normally for 30 days after administration. Notably, no detectable damage or any inflammatory lesions were found in major tissues (heart, liver, spleen, lung, kidney, intestine, skin, and muscle) after UV light exposure in both groups, with no significant difference to the control group (Supplementary Fig. S6). These results further demonstrate the safety of folic-AIEgen based image-guided operation in animal models.

**Fluorescence-guided excision of a mammary SLN in rhesus macaque.** Currently, there are a large number of nanomaterials widely designed for surgical operations, but few of them have been comprehensively evaluated in large animals, such as non-human primate. Given that rhesus macaques share many similarities with humans, the preclinical study on rhesus macaques is of vital importance for clinical evaluation and real

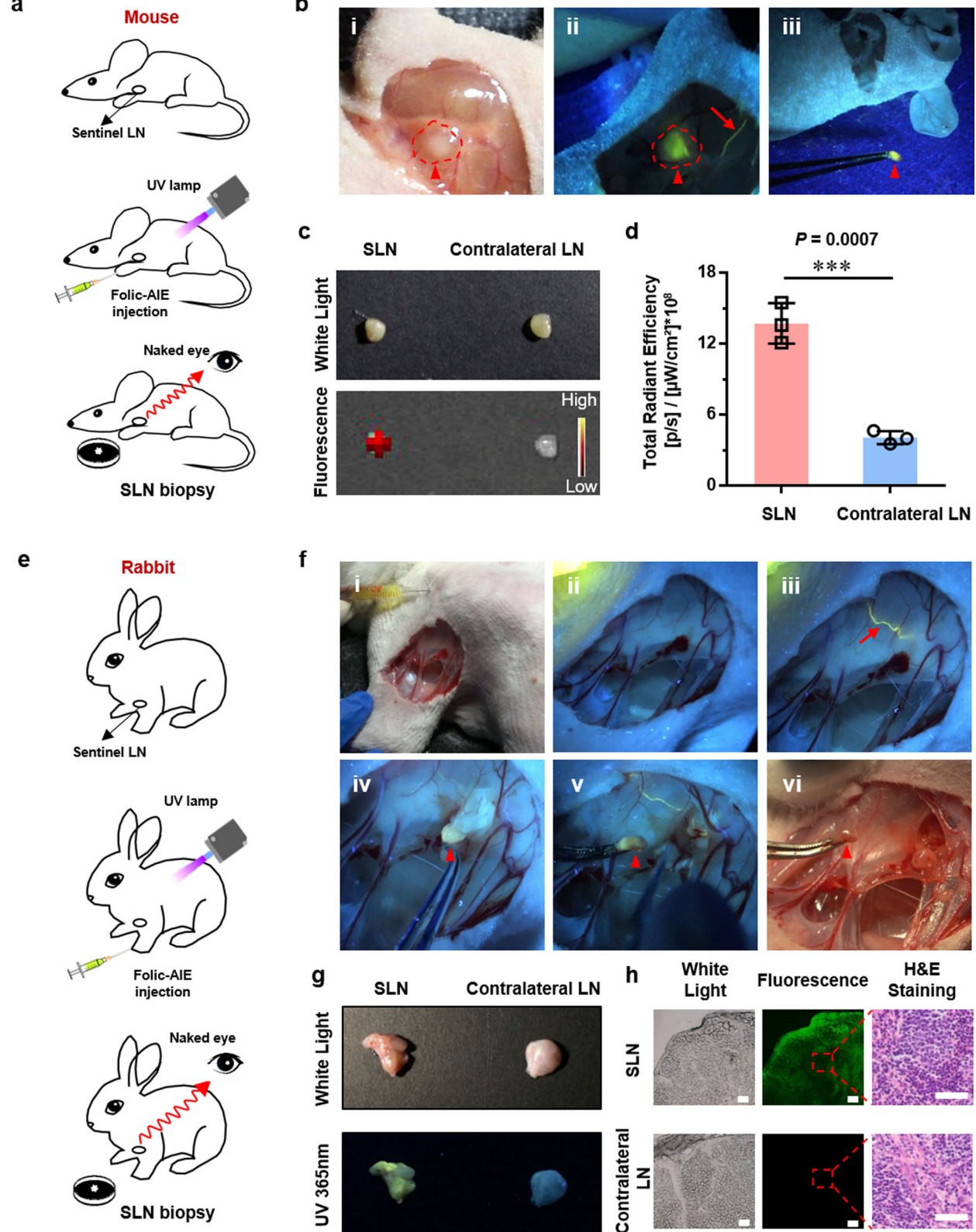

transformation[26]. Notably, rhesus macaque has intricated anatomical structures, including numerous unequal-sized lymph nodes within the axillary fossa, which is entirely different from those of mouse and rabbit[27]. Therefore, folic-AIEgen was further investigated in rhesus macaque's breast SLNs biopsy for evaluating the feasibility and repeatability in fluorescence imaging-guided surgery (Fig. 3a). The lymph node excision was conducted after a subcutaneous injection of folic-AIEgen into the right areola of rhesus macaque's breast (Fig. 3b, i&ii). As shown in Fig. 3b and Supplementary Video 2, folic-AIEgen entered into the lymphatic system and the bright fluorescence could be observed in the lymphatic vessel at about two mins postinjection (Fig. 3b, iii). Particularly, folic-AIEgen moved rapidly and stayed in the SLN within 15 mins, without diffusion to the periphery or

**Fig. 2 Effective SLN biopsy in nude mice and rabbit breast via folic-AIEgen based image-guided operation. a** Representative process of folic-AIEgen guided LN dissection in nude mice ($n = 3$). **b** Images of the lymphatic vessels (red arrows) and SLNs (red arrowheads) under white light and a handheld UV lamp. i-Before injection, ii-30 seconds after injection of folic-AIEgen (25 μL, 40 μg/mL) into the front footpad of mice, iii-Dissection of the LN. **c** Photographs of draining LN and contralateral LN obtained from IVIS system and digital camera. **d** Quantitative analysis of the fluorescence intensity of draining LNs and contralateral LN. Data are means ± SD, $n = 3$, Student's two-tailed t test, ***$P < 0.001$. **e** Representative process of folic-AIEgen guided SLNs dissection from the armpit of rabbit ($n = 3$). **f** Images of the lymphatic vessels (red arrows) and SLNs (red arrowheads) under white light and a handheld UV lamp. i and ii-Before injection, iii-3 min after injection of folic-AIEgen (500 μL, 40 μg/mL) around the second papilla of rabbit, iv-Separation along the lymph vessel, v and vi- Dissection of the SLN. **g** Photographs of draining LN and contralateral LN obtained from the digital camera under white and UV light. **h** Histological analysis of the draining LN and contralateral LN. Representative bright field and photoluminescence images at 488 nm excitation of frozen sections. Histological changes were evaluated by H&E staining. Scale bars = 100 μm. The experiment was repeated at least three times.

second-tier nodes (Fig. 3b, iv). Compared with most SLNs tracers, folic-AIEgen demonstrated a longer retention time within SLNs, which efficiently improved the accuracy and repeatability of SLNs biopsy[31,32]. Additionally, the time interval offered by folic-AIEgen also provided sufficient time for surgeons to localize and remove the LNs via fluorescence signal.

The SLN resection was performed carefully under the guidance of the yellow fluorescence excited by the UV lamp and the dim white light (Fig. 3b, v). After imaging-guided excision, the vanished fluorescence signal clearly indicated that SLN was successfully removed from the regional lymph node basin (Fig. 3b, vi–viii). Correspondingly, strong fluorescent signals could be observed in the harvested SLN, while no apparent fluorescence was detected within adjacent LN (Fig. 3c–e). Consistent with the above results, folic-AIEgen did not induce any detectable morphologic changes in SLN, demonstrating a good bio-compatibility (Fig. 3e).

After the surgery, no abnormalities were observed in the rhesus macaque's behaviors including eating, drinking, urination, defecation, sleeping, activity, grooming, and neurological status, suggesting minimal systemic effects (Supplementary Table 1). The body weight of rhesus macaque was monitored 1-day before and 180-day after the surgical operation. During the whole period, no greater fluctuations were observed, from 6.55 kg before treatment to 5.80 kg for 180 days after treatment (Supplementary Table 2). As the nano-scaled folic-AIEgen has a similar size to viruses and large proteins that might potentially stimulate the immune system and induce inflammatory response[33], standard hematological and biochemical markers were further examined. It could be seen from the complete blood count results that the cells' values were within the normal range for three months post-treatment (Supplementary Fig. S7). Similarly, biochemistry assays including various indicators (e.g., alanine transaminase, aspartate transaminase, urea, creatinine of liver and kidney function) fluctuated within the normal ranges throughout the examination period[24]. All these results clearly demonstrate the healthy condition of animals after folic-AIEgen based surgical operation, indicating the outstanding bio-compatibility of folic-AIEgen.

**Precision image-guided biopsy for breast cancer-metastasized SLNs in nude mice supported by folic-AIEgen.** As one well-studied breast cancer cell line, 4T1 cells can spontaneously metastasize from the primary tumor to multiple organs, including LNs, making it ideal for evaluating the efficiency and accuracy of imaging-guided probes[24]. More importantly, the axillary area of nude mice contains two LNs (proper axillary, accessory axillary nodes), which is similar to human subscapular LNs and central LNs. Precise detection of lymphatic metastasis of breast cancer in nude mice would greatly help address critical clinical trials. Therefore, folic-AIEgen based fluorescene imaging-guided surgery in nude mice bearing 4T1 tumor was conducted (Fig. 4a). As shown in Fig. 4b (i&ii), the lymphatic vessels were gradually brightened after 3 m of subcutaneous injection. The targeted

SLNs were then brightened and carefully removed in a real-time imaging-guided manner (Fig. 4b, iii-v, and Supplementary Video 3). It could be clearly seen that folic-AIEgen particles were mainly distributed in SLN, with very few being in adjacent LN (Fig. 4b, vi, and Fig. 4c). Correspondingly, the metastasis of 4T1 cancer cells was only observed within SLN, showing the high targeting efficiency of folic-AIEgen for fluorescence-guided pre-cision surgery (Fig. 4d). Sentinel lymph node surgery has been widely applied as a standard technique for node-negative breast cancer patients to avoid complete axillary lymph node dissection. However, it usually triggers various sequels including upper-extremity lymphedema, arm numbness, limited shoulder mobi-lity, post-operative infection, etc.[34]. Thus, this folic-AIEgen probe would be a potentially powerful tool for improving the detection rate and accuracy of SLNs biopsy in clinical trials.

**Tumors-specific accumulation of folic-AIEgen in nude mice intraperitoneal xenograft model.** Peritoneal tumors account for approximately 250,000 new cancer cases annually in the USA and result in high mortality rates among patients[35]. An efficient preoperative diagnosis would effectively help surgeons identify malignant tissues and shorten the surgical time. In order to investigate the possibility of applying folic-AIEgen for the pre-operative diagnosis of peritoneal tumors, we further studied the tumor-targeting ability of folic-AIEgen to intraperitoneal tumors in the SKOV3 xenograft mouse model. The fluorescence signal of folic-AIEgen accumulated in the peritoneal tumors of mice was significantly stronger than that of AIEgen without folic acid modification after intraperitoneal (i.p.) administration (Supple-mentary Fig. S8). Furthermore, the ex vivo fluorescence images of intestinal tissues with tumors showed that the folic-AIEgen treatment group's fluorescence signal was significant, mainly distributed in the tumors. In contrast, the signal of AIEgen-treated group was much weaker and randomly distributed in either tumor or adjacent normal tissues. The above illustrates the specific targeting ability of folic-AIEgen for SKOV3 tumors, and then we evaluated the possibility of applying it to image-guided surgery to remove tiny tumors.

First of all, consistent with the above results, there was no difference in histological and blood test results between the control group and the folic-AIEgen group, indicating the high bio-compatibility of folic-AIEgen (Supplementary Fig. S9, 10). The obvious fluorescence signal of the SKOV3 malignant tissues was detected within 6 h after intraperitoneal injection of folic-AIEgen (Fig. 5a). More specifically, the tumor-to-normal tissue (T/N) ratio continuously increased 12 h (from 1.0 to 1.7) after injection and reached its peak at 24 h (3.8), indicating a targeted tumor accumulation of folic-AIEgen (Fig. 5b). The high ratio of T/N given by folic-AIEgen yielded a dual benefit of improving image contrast and increasing surgical efficiencies. Notably, several tiny peritoneal metastatic lesions (including smallest nodules at 22.5 mm³) remained fluorescent and were visible to eyes under the UV lamp (365 nm) even at one day after folic-AIEgen administration

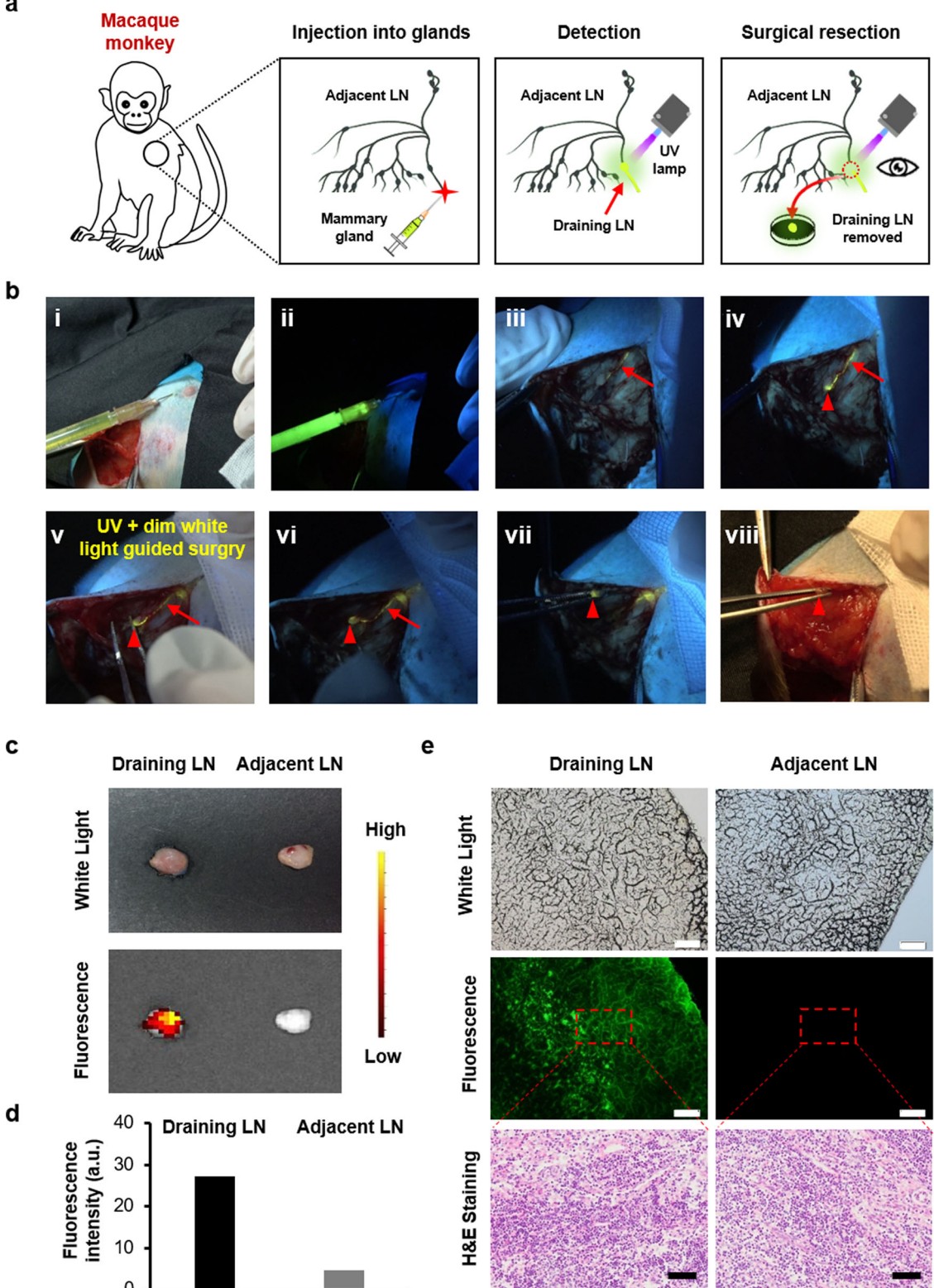

**Fig. 3 Efficient image-guided SLNs biopsy in rhesus macaque breast via folic-AIEgen. a** Representative process of folic-AIEgen guided SLNs dissection from rhesus macaque. **b** Images of the lymphatic vessels (red arrows) and SLNs (red arrowheads) under white light (i, viii) and a handheld UV lamp (ii-vii). i and ii-Before injection, iii-2 min after injection of folic-AIEgen (700 μL, 40 μg/mL) into the right areola of macaque, iv-15 min after injection, v-Separation along the lymph vessel, vi-Exposure of the SLN, vii and viii- Dissection of the SLN. **c** Photographs of draining LN and adjacent LN obtained from digital camera and IVIS system. **d** Quantitative analysis of the fluorescence intensity of draining LN and adjacent LN. **e** Histological analysis of the draining LN and adjacent LN. Representative images of bright field, photoluminescence of frozen section at 488 nm excitation, and H&E staining of paraffin section. Scale bars = 100 μm. The experiment was repeated at least three times.

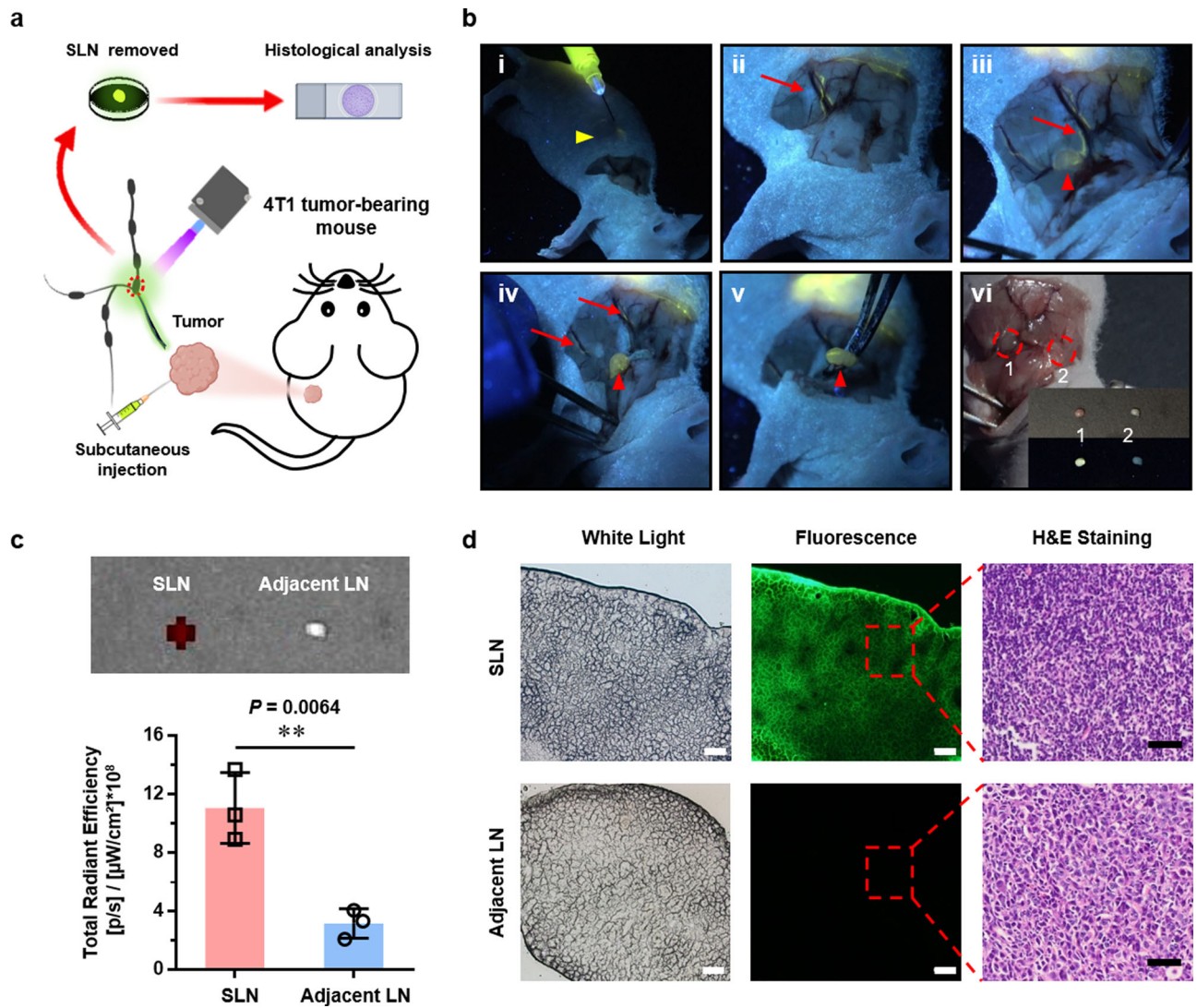

**Fig. 4 Image-guided biopsy for lymph node metastasis of breast cancer in nude mice via folic-AIEgen. a** Representative process of folic-AIEgen guided SLNs dissection in 4T1 breast cancer-bearing mice (*n* = 3). **b** Images of the lymphatic vessels (red arrows) and SLNs (red arrowheads) under a handheld UV lamp (i-v), and white light (vi). i-Injection, ii-3 min after subcutaneous injection of folic-AIEgen (50 µL, 40 µg/mL) around the tumor, iii-Separation along the lymph vessel, iv-Exposure of the SLN, v-Dissection of the SLN. vi. 1 and 2 represent the sentinel LNs and adjacent LN, respectively. The inner picture was obtained in bright field (top) and UV irradiation (bottom). **c** Images were obtained by IVIS system, and the quantitative analysis of fluorescence signals was performed between SLNs and adjacent LNs. Data are means ± SD, *n* = 3, Student's two-tailed *t* test, **P < 0.01. **d** Histological analysis of the LNs. Representative images of bright field and photoluminescence of frozen section at 488 nm excitation, and H&E staining of paraffin section. Scale bars = 100 µm. The experiment was repeated at least three times.

(Fig. 5c). Furthermore, tumors ≤1 mm in diameter could be precisely removed by folic-AIEgen-mediated surgical resection (Fig. 5c). Ideally, chemotherapy could effectively kill residual small tumor nodules (maximum diameter ≤1 cm) in ovarian cancer patients after surgical cytoreduction[36–38]. Therefore, folic-AIEgen assisted cytoreductive surgery would change disease progression rates and thereby prolong survival, like other reported fluorescent probes[3,39].

**Efficient image-guided surgery for tiny, peritoneal-disseminated SKOV3 and Hela tumors via folic-AIEgen.** Unlike systematic administration, the local injection could reduce unnecessary systemic side-effects and further increase the local drug concentration that failed to be reached via the systemic route[40]. Next, folic-AIEgen was administered intraperitoneally to guide the surgery of tiny, peritoneal-disseminated SKOV3 (Fig. 6a) and Hela (Fig. 7a) tumors. All SKOV3 (Fig. 6b) and Hela

(Fig. 7b) peritoneal metastatic tumors were clearly visible under 365 nm UV lamp at 24 h after intraperitoneal injection of folic-AIEgen.

Importantly, SKOV3 and Hela peritoneal metastatic tissues, including tiny lesions (≤1 mm in diameter) could be accurately identified by folic-AIEgen based fluorescence imaging, which was consistent with the tumor locations mapped by firefly luciferase imaging (Figs. 6b and 7b). In addition, the tiny metastatic Hela tumor in the liver could also be detected, showing the high sensitivity of folic-AIEgen (Supplementary Fig. S11). This would make it more convenient for surgeons to perform image-guided surgery and fluorescence-guided laparoscopic biopsy in clinical practice. Meanwhile, the relatively lower dose required in local administration might reduce the potential toxicity induced by the drugs used in molecular imaging[41,42].

Finally, there was no recurrence one week after the surgery and the survival rates of mice were greatly improved, attributing to the

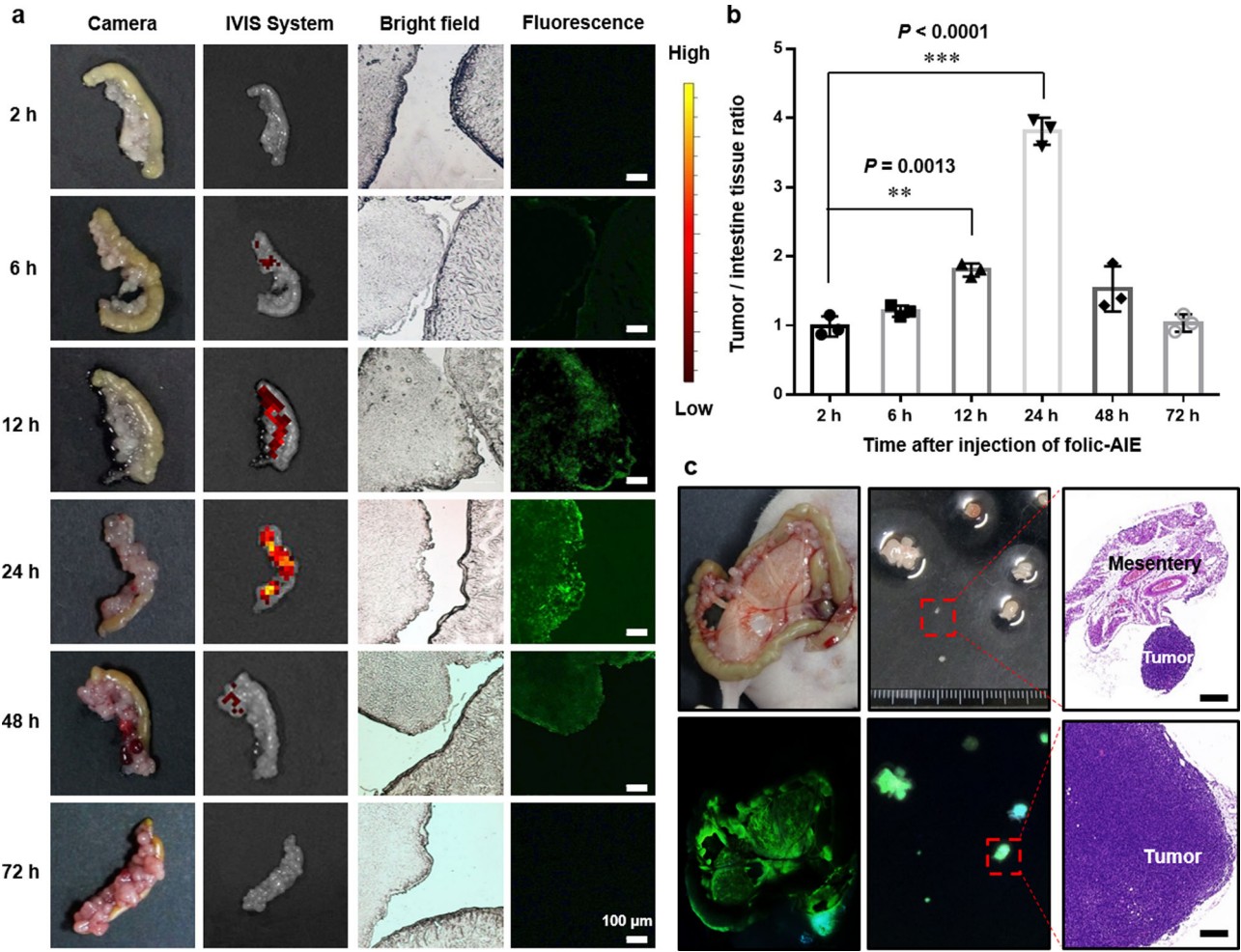

**Fig. 5 Tumors-specific accumulation of folic-AIEgen in intraperitoneal xenograft mouse model. a** Time-dependent accumulation of folic-AIEgen in the SKOV3 tumors. At different time points after intraperitoneal injection of folic-AIEgen (100 µL, 40 µg/mL), mice were sacrificed and opened the abdominal cavities. A lot of tumor nodules could be seen on the surface of intestines, and a section of intestinal tract adherent with tumor nodules was randomly resected from each mouse and imaged using the digital camera, IVIS system, and microscope (bright field and fluorescence at 488 nm excitation), respectively. No obvious fluorescence signal was detected in the tumor site at 2 h postinjection. A steady increase of the fluorescence intensity in SKOV3 tumors was observed in the following 6, 12, and 24 h, and the fluorescence signal decreased at 48 h and disappeared at 72 h. Scale bars = 100 µm. **b** The tumor-to-normal tissue (T/N) ratio in tumor nodules with adjacent intestinal tissues at different time points after folic-AIEgen injection. Data are means ± SD, $n = 3$, Student's two-tailed $t$ test, **$P < 0.01$, ***$P < 0.001$. **c** Images of the mouse peritoneum and excised tumors captured under white light (top) and 365 nm UV light (bottom). Representative H&E staining of the excised tumors. Scale bars = 1 mm. The experiment was repeated at least three times.

complete resection of SKOV3 and Hela malignant tissues via image-guided resection (Supplementary Fig. S12, 13). These results suggested folic-AIEgen can serve as an effective agent for image-guided surgery and shows huge potential in clinical translation.

**Systematic comparison of folic-AIEgen and potential-clinical probes.** Preclinical imaging mediated by PET, MRI, and CT allows the accurate localization and definition of the cancer lesions. However, in most surgical operations, surgeons have to rely on personal experience to identify malignant tissues, which may cause incomplete tumor resection and potentially increase the tumor recurrence rate. Currently, the FDA-approved NIR probe ICG (emit at 800 nm) is under investigation in clinical IGS (e.g., the detection of solid tumors) to promote surgery efficiency[43]. Nevertheless, the application of ICG suffers from the low dispersibility, bad fluorescent stability, and high medical cost in equipment, etc. (Supplementary Fig. S14, 15, and Table 3). In

comparison, the well-dispersed and uniformed (ca 20 nm) AIE-gens provides real-time visualization and accurate SLN detection for naked-eye IGS under a portable UV lamp and reduces the cost in training and equipment, showing great potential as a probe for IGS.

Given that the metastatic cancer cells tend to spread via lymphatic flow, the mapping of the first-chain node, sentinel lymph node, is vital for diagnosis[44]. Currently, ICG is used as NIR lymphatic tracers to map deep-seated lymph nodes, including SLN, due to its enhanced tissue depth penetration (Supplementary Table 4). As a supplement, an intraoperative usage of staining dye (e.g., Evans Blue) is highly demanded to visually identify the sentinel lymph nodes from afferent lymphatic channels, which could improve the dissection accuracy[45]. However, the bleeding during operation would quickly dilute and fade the optical vision generated by Evan blue. Alternatively, radiotracer, such as technetium-99m-labeled nanocolloid (e.g., SiNPs) has also been employed for detecting SLN. In considera-tion of the large size (normally larger than 50 nm), slow clearance,

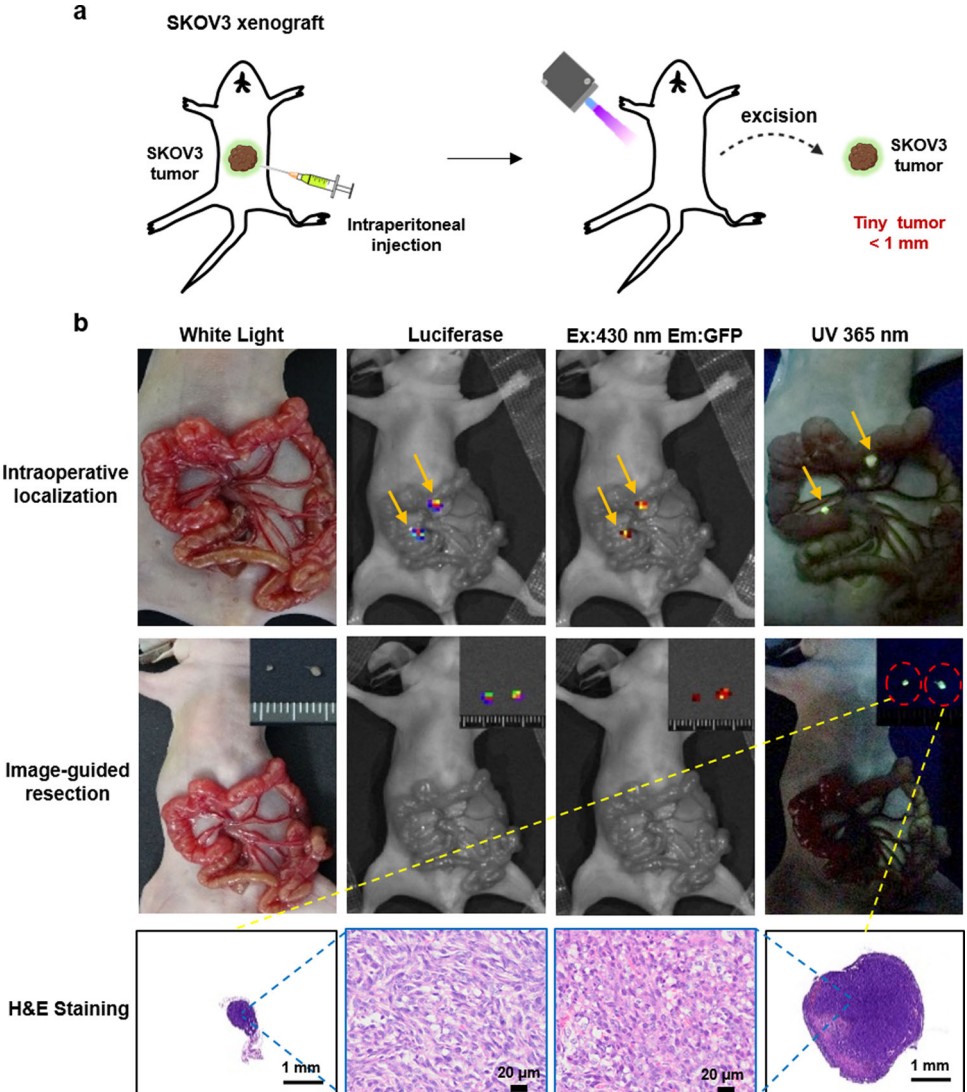

**Fig. 6 Image-guided surgery for small, peritoneal-disseminated SKOV3 tumors. a** Representative process of folic-AIEgen guided dissection of tiny tumors in SKOV3-Luc xenograft-bearing nude mice ($n = 3$). At 24 h after intraperitoneal injection of folic-AIEgen (100 µL, 40 µg/mL), the intestinal tract of mice was carefully pulled out and the peritoneal SKOV3 microtumors were localized and resected under UV light. **b** Tumor co-localization (top) by the bioluminescence signals of SKOV3-Luc, fluorescence of folic-AIEgen. Image-guide tumor resection (middle) via the fluorescence of folic-AIEgen under 365 nm UV light (inner: the excised tumors). Bioluminescence imaging was performed on the mice again to confirm no residual lesions. H&E staining of the excised tumors (bottom). Scale bars = 1 mm. The experiment was repeated at least three times.

and radioactivity, the application of nanocolloid for SLN mapping is still under contention[46]. In this study, the high biocompatible and sensitive folic-AIEgen was synthesized, with a suitable size around 20 nm, which could help folic-AIEgen escape from a quick clearance via either kidney (<10 nm; e.g., ICG) or liver/spleen (>100 nm; e.g., SiNPs and superparamagnetic iron oxide nanoparticles[47]) (Supplementary Fig. S16). This folic-AIEgen provided a naked-eye visualization of SLN in real-time and ensured efficient and successful IGS (Figs. 2, 3).

As one common issue among potential-clinical IGS probes (e.g., ICG, methylene, blue and 5-aminolevulinic acid), the non-targeting feature inevitably generates a high background signal that eventually hinders these probes from precision IGS of malignant lesions[48]. In contrast, the conjugation of targeting ligand, molecule, or antibody on nano-probe could enhance tumor tissues' specific affinity during the surgery[49,50]. Similarly, folic-modified AIEgen is expected to achieve specific tumor targeting and provide an enhanced intratumoral fluorescence

signal for IGS (Figs. 4–7). More importantly, the AIE signal will only be generated via the accumulation that normally happens inside the tissues, such as tumors. In comparison with most conventional luminescent materials that generally suffer from aggregation-caused quenching (ACQ), this feature strongly reduces the background signal caused by diffusion and further enhances the AIE signal within the tumor. For instance, the folic-AIEgen was able to detect tiny malignant lesions less than 10 mm$^3$, while the ICG-loaded TMTP1-PEG-PLGA micelles could only detect cancerous nodules above 90 mm$^3$ (Figs. 5c, 6b and 7b)[51–54].

It is notable that a facile and repeatable synthesis approach is highly desired during the clinical translation the IGS probes. Unlike other multifunctionalized nanoprobes that require cumbersome synthesis steps, folic-AIEgen could be prepared via simple chemical methods[28,29]. Additionally, folic acid is much more cost-effective than other targeting molecules, such as tumor-targeting ligand and aptamer, which could greatly lower

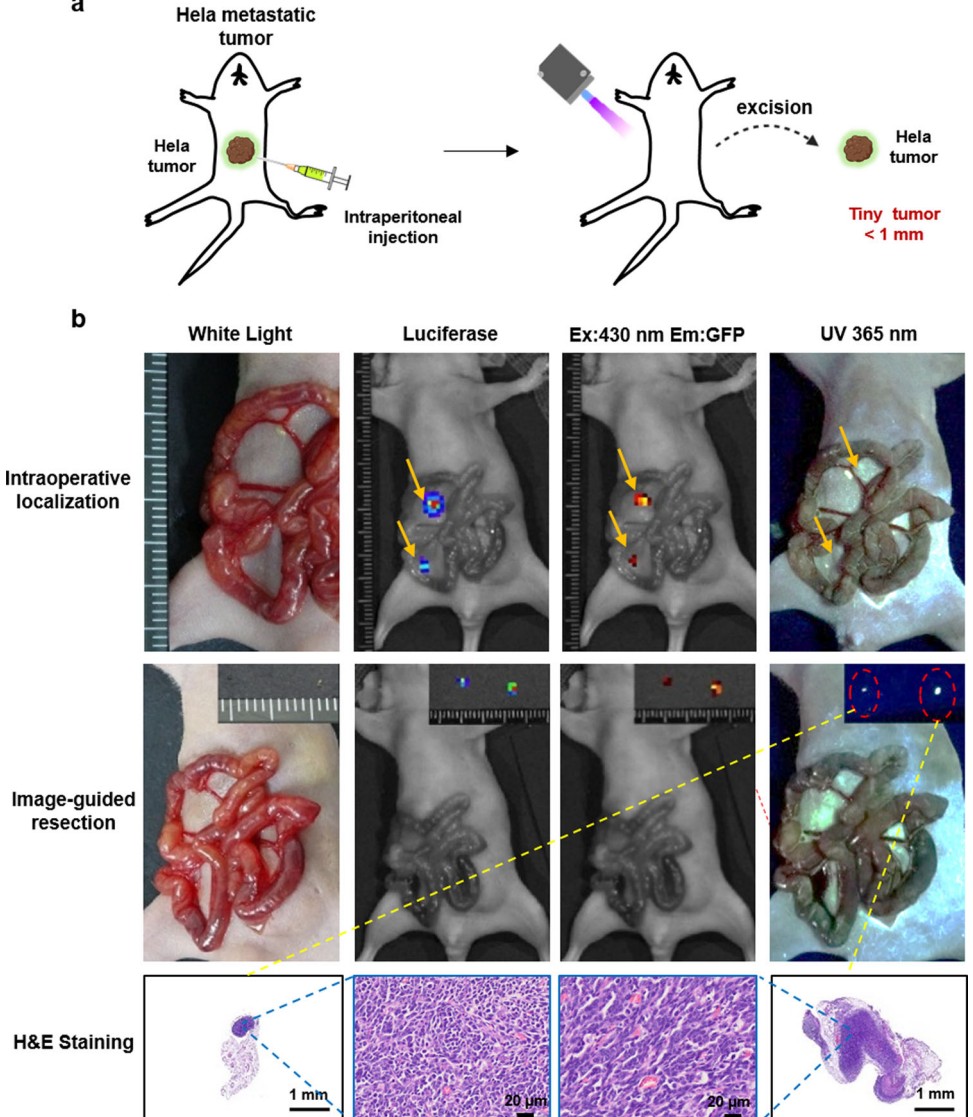

**Fig. 7 Image-guided surgery for small, peritoneal-disseminated metastatic Hela tumors. a** Representative process of folic-AIEgen guided dissection of tiny tumors in Hela-Luc xenograft-bearing nude mice ($n = 3$). At 24 h after intraperitoneal injection of folic-AIEgen (100 μL, 40 μg/mL), the intestinal tract of mice was carefully pulled out and the peritoneal Hela microtumors were localized and resected under UV light. **b** Tumor co-localization (top) by the bioluminescence signals of Hela-Luc, fluorescence of folic-AIEgen. Image-guide tumor resection (middle) via the fluorescence of folic-AIEgen under 365 nm UV light (inner: the excised tumors). Bioluminescence imaging was performed on the mice again to confirm no residual lesions. H&E staining of the excised tumors (bottom). Scale bars = 1 mm. The experiment was repeated at least three times.

the cost during synthesis[49,50]. More importantly, these two widely-studied elements, folic acid, and AIEgens, have demonstrated their safety in preclinical studies, which will hopefully lead to advances in quality control (QC) and future clinical translation.

## Discussion

Here, we report the facile preparation of folic-AIEgen and its multifunctional applications in image-guided surgery. Take advantage of stable fluorescence and suitable size, folic-AIEgen have been successfully employed in sensitive detection and precise dissection of sentinel LNs from mice to nonhuman primate rhesus macaque. Furthermore, our results suggest that this specific targeting probe is capable of performing rapid tumor accumulation and superior T/N ratio, which helps detect tiny peritoneal-disseminated tumors and guide tumor resection, significantly ensuring surgical success and reducing the tumor

recurrence rate. Systematic biosafety assessment in a variety of animal models further proves the excellent bio-compatibility of folic-AIEgen. Notably, AIEgens has been successfully applied in the rhesus macaque model in this study. Our work has provided valuable data for preclinical studies in nonhuman primate, which would greatly facilitate further progress of folic-AIEgen and other AIE-based luminescence probes in clinical translations.

## Methods

**Preparation of folic-AIEgen.** To prepare AIEgen (BTBEBT)[28], 1 mL of THF solution containing BTPEBT (1 mg) and DSPE-PEG$_{2000}$ (1.5 mg) was added into 9 mL of MilliQ water under ultrasound sonication. The mixture was stirred at room temperature overnight, purified by ultrafiltration, and finally filtered by a 0.22 μm filter. To prepare folic-AIEgen[29], 300 μL of BTBEBT THF solution (3 mg/mL), 300 μL of DSPE-PEG$_{2000}$ THF solution (3 mg/mL), and 300 μL of DSPE-PEG$_{2000}$-folic THF solution (3 mg/mL) were mixed together. The mixture was then added dropwisely into 9 mL of MilliQ water under ultrasound sonication. THF in the mixture was removed by N$_2$ and dialyzed via a membrane with a molecular weight cutoff of 3500 g/mL. At last, the solution was filtered by a 0.22 μm filter.

**Characterization of folic-AIEgen**. Morphology of folic-AIEgen was characterized on a FEI Tecnai F20 transmission electron microscopy (Hillsboro, OR, USA). Ultraviolet–visible–near-infrared (UV-vis-NIR) spectra of folic-AIEgen were recorded on a Shimadzu 2600 UV-vis-NIR spectrophotometer (Shimadzu, Kyoto, Japan). The hydrodynamic size was measured on a Malvern Zetasizer Nano-ZSE (Malvern, UK) at 25 °C.

**Cell culture and luciferase transfection**. SKOV3 human ovarian carcinoma cells (ATCC, VA) were cultured in McCoy's 5 A medium. HaCaT human keratinocyte cells (ATCC, VA) and Hela colon carcinoma cells (ATCC, VA) were cultured in dulbecco's modified eagle medium (DMEM). 4T1 murine breast cancer cells (ATCC, VA) were cultured in RPMI-1640 medium. All mediums were supplemented with 10% fetal bovine serum (FBS) and 1% antibiotics (100 U/mL penicillin and 100 μg/mL streptomycin), and all cells were incubated at 37 °C with 5% $CO_2$. To establish SKOV3/Luc and Hela/Luc stable cell lines, exponentially growing SKOV3 and Hela cells at 30% confluency were infected with lentivirus (Ubi-MCS-firefly_Luciferase-IRES-Puromycin, GeneChem, Shanghai, China) according to the product guideline. Then positive cells after transfection were screened and harvested by puromycin supplement for future assays.

**Cytotoxicity assays of folic-AIEgen**. HEK293 human embryonic kidney cells (ATCC, VA), HL7702 human hepatocyte cells (ATCC, VA), human fibroblast cells (ATCC, VA), and 4T1 cells were seeded in 96-well plates with 5,000 cells per well and cultured overnight. The cells were refreshed with mediums containing a series concentrations (0~100 μg/mL) of folic-AIEgen and incubated for 24 h. Then, the viabilities of cells after treatments were analyzed via MTT assay kit (YEASEN, Shanghai, China) and measured on a plate reader (SpectraMax M5, Molecular Devices, CA, USA).

**Folate receptor-mediated uptake of folic-AIEgen**. To determine the folate receptor-depended uptake of folic-AIEgen, HaCaT, and SKOV3 cells were incubated with 20 μg/mL folic-AIEgen for 24 h. The folate receptor-induced endocytosis of folic-AIEgen was further confirmed by blocking the receptor with 1 mmol/L free folate folic acid before the addition of folic-AIEgen. After incubation, cells were washed with PBS, fixed with 4% polyformaldehyde, and then stained with DAPI. Images were captured by an inverted fluorescence microscope (Olympus IX71, Japan).

**Animal models**. All animal procedures were performed according to the National Institute of Health Guide for the Care and Use of Laboratory Animals. All animal (including mice, rabbits, and monkeys) studies were approved by the Institutional Animal Care and Use Committee of Zhejiang University. New Zealand rabbits (female, weighted 3.0 kg) and BALB/c nude mice (female, 6 weeks old) were obtained from Zhejiang Chinese Medical University Laboratory Animal Research Center. Rhesus macaque (female, 5 years old and weighed 6.0 kg) was bought from Soochow Xishan Zhongke Lab Animal Co. Ltd, Suzhou, China. Mice were acclimated to the housing conditions in an ambient temperature (21 °C) and humidity (55%), with 12/12 h light/dark cycles, and free access to food and water ad libitum. The 4T1 tumor model was established by subcutaneous (s.c.) injection of 50 μL of 4T1 cell suspension (approximately $5 \times 10^6$ cells) into the flank regions of BALB/c nude mice ($n = 3$). To establish the intraperitoneal xenograft model, SKOV3/Luc and Hela/Luc (approximately $3 \times 10^6$ cells) cells were injected into the abdominal cavities of BALB/c nude mice ($n = 3$).

**Image-guided LN biopsy in nude mice**. To identify the SLNs site, BALB/c nude mice ($n = 3$) were subcutaneously injected with folic-AIEgen (25 μL, 40 μg/mL) into the front footpad under anesthesia with 1–2% (vol/vol) isoflurane in $O_2$. The skin around the LNs region was removed preoperatively to observe the folic-AIEgen fluorescence from the injection site to LNs. Image-guided surgery of SLNs, including contralateral axillary and adjacent LNs was performed under the irradiation (365 nm) of a handheld UV lamp and recorded with a digital camera. The lymph nodes were harvested for imaging and histological analysis. The UV light source is a 365 nm UV LED (PKG LEUVA35T01RL00, LG, Korea), which has a UV intensity of 10 mW/cm$^2$ at a distance of 25 cm.

**Image-guided biopsy of breast SLNs in rabbit**. The rabbits were anesthetized with isoflurane using a face mask during the experiment. The hair around the right armpit and breast was shaved and a small skin incision was made on the right armpit for dynamic movement tracking of folic-AIEgen ($n = 3$). After subcutaneous injection of folic-AIEgen (500 μL, 40 μg/mL) around the second papilla on the right prothorax, the image-guided surgery was performed under the 365 nm irradiation of UV lamp, and the procedure was recorded with a digital video camera. Subsequently, the SLNs and contralateral LNs were collected for imaging and histological assays. The skin incisions were sutured after SLNs biopsy, and animals were treated with cefazolin by intramuscular (i.m.) injection every 24 h for 3 days.

**Image-guided biopsy of breast SLNs in rhesus macaque**. The animal subject was determined to be normal and active during the preoperative assessment conducted the day prior to surgery. Food was withheld for 8 h prior to surgery and water was withheld for 2 h prior to surgery. General anesthesia was induced with ketamine HCl (10 mg/Kg, IM) with atropine (0.03 mg/Kg, IM). Once anesthetized, the animal was endotracheally intubated and anesthesia was maintained with 1–2% isoflurane mixed with 100% oxygen. Heart rate, body temperature, oxygen saturation (SpO$_2$), and end-tidal $CO_2$ (ETCO$_2$) were monitored by a physiological monitor (iPM 12 Vet, Mindray, China). The hair from the right axilla was clipped and the axillary region was sterile with povidone iodine for surgery using. Next, a 5 cm transverse incision was made in the right axilla to visualize the fat pad and lymph nodes region. Folic-AIEgen (700 μL, 40 μg/mL) was then injected into the mammary tissue of the right areola. Using a 365 nm UV light was directed to the incision site and the progression of the fluorescent indicator through the lymph vessels and into the draining lymph node (or SLN) was recorded with a digital camera. The once illuminated and identified by the surgeon, the SLN was excised. The skin incision was then sutured with 3-0 nylon vertical mattress sutures and a bandage was applied. Maintenance anesthesia was discontinued, the animal was recovered from anesthesia and returned to the home cage. A prophylactic antibiotic (ceftriaxone sodium, 1 g daily, IM) was administered for 5 days after surgery. Analgesia was provided with buprenorphine (0.06 mg/Kg) administered pre-op and then every 8 h for 3 days post-op. Post-operative animal behavior, appetite, body weight, and blood work were monitored and reported as normal.

**Image-guided SLNs biopsy in 4T1 tumor-bearing nude mice**. SLNs biopsies were performed on 4T1 tumor-bearing BALB/c nude mice ($n = 3$) with tumors of approximately 1.0 cm$^3$. Mice were anesthetized with 1–2% (vol/vol) isoflurane in $O_2$, and the skin of axilla was removed for tracking the fluorescence of folic-AIEgen. The image-guided surgery was performed under a handheld UV lamp, after subcutaneous administration of folic-AIEgen (50 μL, 40 μg/mL) around the tumor. At the end of experiment, animals were euthanized and lymph nodes were harvested for imaging and histological analysis.

**Targeted imaging of intraperitoneal SKOV3 xenograft in nude mice**. The tumor growth of intraperitoneal SKOV3/Luc xenograft was regularly monitored by luciferase activity. Briefly, tumor-bearing nude mice were intraperitoneally (i.p.) injected with D-luciferin (10 μg/g body weight). Bioluminescence images of mice were collected using IVIS Lumina LT Series III (Perkin Elmer, USA) at 10 mins postinjection. The bioluminescence signals were quantified using the Living Image 4.3.1 software. Mice bearing SKOV3 xenografts with bioluminescence signals greater than $1 \times 10^9$ photon/sec/cm$^2$/sr were selected for the experiment. To investigate the time-dependent tumor accumulation of folic-AIEgen, the intestine tissues with tumors were collected from SKOV3 bearing mice ($n = 3$) for imaging and histological assays at different time points (2, 6, 12, 24, 48, and 72 h post i.p. injection of folic-AIEgen).

Based on the time-dependent distribution of folic-AIEgen, image-guided surgeries of SKOV3 malignant tissues were performed at 24 h post i.p. injection of folic-AIEgen. Following anesthesia (1–2% (vol/vol) isoflurane in $O_2$), the abdominal cavity of mice ($n = 3$) was opened, and the intestines were carefully pulled out. With the assistance of 365 nm UV lamp, large tumors and small eye-invisible lesions in the peritoneum were carefully excised under fluorescence guidance. All tissues were collected for further imaging and histological assays.

**Image-guided surgery for tiny, peritoneal-disseminated SKOV3 and Hela tumors**. Image-guided surgery were performed on both SKOV3/Luc and Hela/Luc xenograft models in mice ($n = 3$), when the firefly luciferase activity signal obtained by the IVIS system reached about $3 \times 10^8$ photon/sec/cm$^2$/sr. at 24 h after intra-peritoneal administration of folic-AIEgen (100 μL, 40 μg/mL), malignant tissues and peritoneal tumors were identified by bioluminescent signals and AIE signals (with excitation at 430 nm) obtained by IVIS Lumina LT Series III (Perkin Elmer, USA), as well as visible fluorescence under UV lamps. All positive tissues were carefully excised under UV light (365 nm) and further analyzed by histochemical staining. After resection, the mice were regularly monitored for tumor recurrence by bioluminescence for up to 7 days.

**Statistical analysis**. All data are expressed as means ± SD, and the number of samples is indicated in each figure legend. Results are representatives of at least three independent experiments. The Student's $t$ test was used for comparisons between two groups of experiments. Statistically significant $P$ values are indicated in Figures and/or legends as ***$P < 0.001$; **$P < 0.01$; *$P < 0.05$. Microsoft Excel 2016 and GraphPad Prism 8 was used to analyze the data in this study.

**Reporting Summary**. Further information on research design is available in the Nature Research Reporting Summary linked to this article.

## Data availability

The data supporting the findings from this study are available within the article and its supplementary information. Source data are provided with this paper and available on

figshare at https://figshare.com/s/2ced8cf4764c8a08d9f3. Any remaining raw data will be available from the corresponding author upon reasonable request. Source data are provided with this paper.

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

## Acknowledgements

This work was supported by the National Natural Science Foundation of China (21788102, 81725009, and 81971667), the Key Research and Development Project of Zhejiang Province (No. 2020C03035), and the Fundamental Research Funds for the Zhejiang Provincial Universities (No. 2021XZZX034). The authors are grateful for the technical support by the Core Facility, Zhejiang University School of Medicine.

## Author contributions

B.Z.T., M.T., and M.Z. designed the project. D.Z., W.C., Z.X., R.H., Y.Q., B.Z., W.L., J.H., and Z.W., performed the experiments. D.Z., Z.X., B.Z.T., Z.Z., and M.Z. analysed and interpreted the data. B.Z.T. and M.Z. supervised the overall research. D.Z. W.C., D.D., B.Z.T., and M.Z. wrote the manuscript.

## Competing interests

The authors declare no competing interests.
