## [Peer Review File · Nature Communications]

REVIEWER COMMENTS

Reviewer #1 (Remarks to the Author):

The manuscript by Zhong et al describes an in-depth study on image-guided surgery in different animal models ranging from mice up to non-human primates using fluorescent nanoparticles (NPs) exhibiting aggregation-induced emission (AIE) behavior. Using already reported formulation of AIE NPs functionalized with folate targeting ligand, it is shown that precise localization and resection of xenograft tumors can be achieved using a simple UV lamp. Overall, this is a very important work showing an attempt to improve precision of the surgical operation using simple and cost-effective techniques. Moreover, the application to non-human primates is a key step forward to clinical translation of these nanomaterials. Therefore, this work could be of interest for a broad community of readers of Nature Communications. However, several major issues and technical points should be addressed before this work can be recommended for publications. Therefore, I recommend major revisions.

Major issues

1) The authors stressed that the use of UV lamp excitation and visible fluorescence light observation is much simpler than the use of near-infrared (NIR) based instruments. However, the author should also clearly explain in the manuscript the limitations of using UV-visible spectral range, which is commonly accepted as less appropriate for biomedical applications. First, it is well known that UV-visible light has much smaller penetration depth in comparison to NIR. In fact, the same authors intensively develop NIR imaging agents and they know that NIR region is much more appropriate for biomedical applications. What is the penetration depth of the technique reported here? Will the limited penetration depth of the UV-visible light will be enough in order to translate this technique to humans? To address these questions, the authors should directly compare their reported system with commonly used ICG contrast agent. Second major problem is the use of UV light, which is well known to be harmful. During the operation, the authors apply UV light directly on the tissues, which are totally unprotected (unlike skin). This may lead to severe complications after the operation. Therefore, an additional study should be made to show that the used doses of UV light are not harmful for the animals.

2) There are also major issues related to folate targeting experiments. First, in vitro experiments show that free folate can block the cell targeting by the AIE NPs. However, in the caption of Figure S4, no information about concentrations of the used free folate and the NPs is provided. In the Methods section, I could find that the used concentration is 1 mM, which is 10,000,000-fold (!) higher compared to the affinity of the receptor to the folate mentioned on page 6 (0.1 nM). At this high concentration, folate can block any sort of binding to the cells surface, so it does not really prove the specificity to the receptor. The second issue is the lack of experimental data in vivo on control NPs without folate ligand. It is hard to evaluate the importance of specific targeting without this control. It should be added at least for the key experiments with xenograft tumors.

3) Scheme 1 is not sufficiently informative and specific. It should be improved and converted into a Figure by including the chemical structure of the dye used, presentation of the nanoparticle and its basic optical characteristics. For the moment, this key information is present in SI, but it would be much better for the reader if this information is presented in a figure of the main text.

Technical issues

1) Important technical information is missing in multiple figure captions: concentration of components used (Figures 1-6) and excitation wavelength used in microscopy experiments (Figures 1-4). Similar problem is with figure captions in the supporting information.

2) Page 6: the size by DLS of 20.3 nm is too precise. Moreover, the error should be provided for the size values presented on this page.

3) Page 7: "A small volume of samples was administrated into the front footpad of nude mice and allowed to move freely." For a broad readership it will not be clear whether the injection was done into a blood circulation or a lymphatic system. Moreover, value of the volume and the total dose of the material injected should be provided here.

4) Methods section (page 21): the protocol for the formulation of folate modified NPs (folic-AIEgen) should be provided.

Reviewer #2 (Remarks to the Author):

The authors ignore the fundamental founding body of literature for fluorescence guided surgery (FGS) which significantly undercuts the novelty of the present manuscript. Please see Reviewer Attachment for an example of this literature. In addition the authors do FGS experiments with luciferase photon counting for FGS, which has no relationship to the clinic since photon counting for FGS in the clinic is not practical.

Robert M. Hoffman

Reviewer #3 (Remarks to the Author):

This manuscript describes research utilizing a novel folic-AIEgen nanoparticles for the intraoperative visualization of lymph vessel, draining lymph nodes, as well as induced carcinomas and their metastases in animal models. The technique was demonstrated in a rhesus macaque by injecting the AIE luminogen into mammary tissue and then following the draining lymph vessels to the illuminated axillary lymph node which was then excised. This technology has great translational value for use in human therapeutic surgical interventions. The authors are correct in noting the surgical time reduction, surgical efficacy, and identification of small metastases may be improved with targeted fluorescent image guidance. All of these improve patient outcomes by reducing surgery time and reducing post-operative metastatic recurrence. I was asked to focus my attention on the rhesus macaque lymph node excision section, although several of my comments apply to the entire document.

The weaknesses of the manuscript:

- 1) More information or references are needed about the relative safety of AIE luminogen when administered into tissues. Specifically, how the agent is metabolized and/or eliminated and how this compares to ICG metabolism and elimination. This is important for patient safety.
- 2) The entire manuscript needs to be edited by a native speaker of English as there are atypical word choices and expressions throughout.

Editorial comments:

- 1) Line 491: recommend changing title to be more specific, "Fluorescence-guided excision of a mammary SLN in rhesus macaque"
- 2) Line 492-512: I've rewritten this section and suggest it read as follows. I made some assumptions about the surgical technique and monitoring parameters, so please check for accuracy. Questions and suggestions are provided in parentheses.

The animal subject was determined to be normal and active during the pre-operative assessment conducted the day prior to surgery. Food was withheld for 8 hours prior to surgery and water was withheld for 2 hours prior to surgery. General anesthesia was induced with ketamine HCl (10 mg/Kg, IM) with atropine (0.03 mg.Kg, IM). Once anesthetized, the animal was endotracheally intubated and anesthesia was maintained with 1-2% isoflurane mixed with 100% oxygen. Heart rate, body temperature, oxygen saturation (SpO₂), and end-tidal CO₂ (ETCO₂) (authors - I think you were monitoring ETCO₂, which is an approximation of PaCO₂. If you were monitoring PaCO₂, please specify the blood gas machine used) were monitored by a physiological monitor (iPM 12 Vet, Mindray, China). The hair from the right axilla was clipped and the axillary region was sterilely prepped for surgery using (authors - specify agent: betadyne? or chloraprep?? - and describe sterile prep here). Next, a 5cm incision (??) was made in the right axilla to visualize the region of the fat pad and lymph nodes. Folic-AIEgen was then injected into the mammary tissue of the right areola. Using a 365 nm UV light was directed to the incision site and the progression of the fluorescent indicator through the lymph vessels and into the draining lymph node (or SLN) was recorded with a digital camera. The once illuminated and identified by the surgeon, the SLN was excised. The skin incision was then sutured (add suture material, gauge, pattern) and a bandage was applied. Maintenance

anesthesia was discontinued, the animal was recovered from anesthesia, and returned to the home cage. A prophylactic antibiotic (cefazolin, 25 mg/Kg IM) was administered at the time of surgery and every 24 hours for 7 days post-op (authors - please check this as cefazolin is normally administered every 8-12 hours rather than every 24 hours). Analgesia was provided with buprenorphine administered pre-op and then every 8 hours for 3 days post-op (authors - please check the dose administered here - the published dose range is 0.01-0.03 mg/Kg every 8-12 hours, which is about 4 times more than you are reporting here. The dose reported would provide little, if any, analgesia). Post-operative animal behavior, appetite, body weight, and blood work were monitored and reported as normal. (reference Fig. S7 here)

Theodore Hobbs, DVM, MCR

Reviewer #4 (Remarks to the Author):

This submission by Zhong et al. reported aggregation-induced emission luminogens for image-guided surgery in multiple animal models including mouse, rabbit and rhesus macaque through targeting folate receptor. The as-prepared folic-AIEgen was characterized with appropriate physicochemical properties and biocompatibility. In vivo toxicity was also determined. The authors first performed image-guided biopsy in mice, rabbits and rhesus macaque which showed visualization of SLNs. Then the authors tested this strategy in multiple tumor models and demonstrated the efficiency of dissecting the tumors and metastatic SLNs, especially in the survival studies. Overall, this is an interesting study based on solid development an image-guided strategy to help the surgical operation. However, there are major concerns associated with the current version of submission which are detailed below.

1. The authors proposed to use folate receptor as a targeting strategy, which is a widely used target for multiple cancer types. However, different subtypes of folate receptor could have different applications in certain tumors. It is necessary to be more specific on this. Moreover, the levels of folate receptor were not characterized in all the tumors which is necessary for this study.
2. One major concern is the lack of demonstration of targeting specificity in this study. There was no control AIEgen tested in the same animal models to validate folate receptor mediated uptake.
3. Folate receptors such as FR β are also overexpressed by myeloid cells and macrophages, which may complicate the operation if using the proposed strategy. Can the authors comment on this?
4. In all the studies, the authors had very few replicates with n=3. It is necessary to increase the rigor of the submission with more replicates.
5. The AIEgen used in this study was approximately 20 nm. There could be significant liver or spleen accumulation. However, it was not observed in Figure S10. It will help the audience to better understand the potential of as-developed strategy if the authors include the biodistribution data following intravenous injection.

6. Local administration was mostly used in the manuscript. However, it would be difficult to do it in clinics. Can the authors comment on this aspect, especially the detection of distant metastasis?

7. There are some typos in the manuscript. For example, in scheme 1, it should be "bedside", not "beside".

Taken together, this is an interesting study and the results have the potential for future translation. Given the major issues discussed above, I would recommend a major revision prior to the acceptance for a publication on Nature Communications.

Reviewer #1:

The manuscript by Zhong et al describes an in-depth study on image-guided surgery in different animal models ranging from mice up to non-human primates using fluorescent nanoparticles (NPs) exhibiting aggregation-induced emission (AIE) behaviour. Using already reported formulation of AIE NPs functionalized with folate targeting ligand, it is shown that precise localization and resection of xenograft tumors can be achieved using a simple UV lamp. Overall, this is a very important work showing an attempt to improve precision of the surgical operation using simple and cost-effective techniques. Moreover, the application to non-human primates is a key step forward to clinical translation of these nanomaterials. Therefore, this work could be of interest for a broad community of readers of Nature Communications. However, several major issues and technical points should be addressed before this work can be recommended for publications. Therefore, I recommend major revisions.

***Comment 1:** The authors stressed that the use of UV lamp excitation and visible fluorescence light observation is much simpler than the use of near-infrared (NIR) based instruments. However, the author should also clearly explain in the manuscript the limitations of using UV-visible spectral range, which is commonly accepted as less appropriate for biomedical applications. First, it is well known that UV-visible light has much smaller penetration depth in comparison to NIR. In fact, the same authors intensively develop NIR imaging agents and they know that NIR region is much more appropriate for biomedical applications. What is the penetration depth of the technique reported here? Will the limited penetration depth of the UV-visible light will be enough in order to translate this technique to humans? To address these questions, the authors should directly compare their reported system with commonly used ICG contrast agent. Second major problem is the use of UV light, which is well known to be harmful. During the operation, the authors apply UV light directly on the tissues, which are totally unprotected (unlike skin). This may lead to severe*

complications after the operation. Therefore, an additional study should be made to show that the used doses of UV light are not harmful for the animals.

Response: Thanks for the reviewer's constructive comment. Indeed, folic-AIEgen fails to act as a contrast agent for primary diagnosis with limited tissue penetration, due to its UV-based imaging approach, as we have clearly shown in **Table 4**. However, this feasible and efficient imaging system shows a great advantage in IGS, which is based on pre-surgical imaging diagnosis. Firstly, folic-AIEgen was subcutaneously (s.c.) injected into the left flanks of BALB/c nude mice, and the fluorescence of folic-AIEgen could be clearly observed under a 365 nm UV light used in our study. This indicates that the UV light source can penetrate the dermis and achieve a penetration depth of about 180 μm [1], which is similar to the human tissue penetration depth of 160 m in the previous study [2]. Although NIR region could achieve deeper penetration, the depth provided by current UV is enough to light up bare blood vessels via our imaging contrast agent during the surgical operation, even in a human trial.

Photographs of BALB/c nude mice under white light and 365 nm UV light after subcutaneous injection with folic-AIEgen. Red arrows indicate the injection site.

Currently, fluorescent imaging application of NIR fluorophores (i.e., ICG) is established for detecting SLN in surgical resection. To clearly address these questions, we systematically compared ICG and folic-AIEgen, including fluorescence characteristics, fluorescence stability and their applications for SLN detection. The detailed description and results are presented on page 13, line 346-353 and Supplementary Fig. 14&15 and Table 3.

Compared with the excitation and emission spectra of ICG in the near-infrared region (780-850 nm) which was invisible to the naked eye, the fluorescence emission of folic-AIEgen at a visible wavelength (~ 540 nm) could be directly observed (Supplementary Fig. 14A&B). In addition, folic-AIEgen exhibited superior fluorescence stability and photostability compared with ICG, which was of great significance for clinical applications. Under physiological conditions (phosphate-buffered saline, 37 °C), ICG showed severe aggregation within 6 h, and the fluorescence intensity decreased sharply, dropping to < 40% within 24 h (Supplementary Fig. 14C&D). In contrast, folic-AIEgen still exhibited stable fluorescence and maintained high transparency after 3-day storage (PBS 37 °C). Furthermore, the fluorescence of ICG gradually reduced under continuous laser irradiation, which decreased to ~52% after 105 min laser irradiation (Supplementary Fig. 14E&F). Comparatively, the fluorescence of folic-AIEgen preserved > 80 % of initial intensity with increased irradiation time.

Supplementary Figure 14. Comparison of fluorescence characteristics and stability between ICG and folic-AIEgen. (A) UV-vis spectra and (B) fluorescence spectra of ICG and folic-AIEgen. (C) Time-dependent photographs (top) and fluorescence imaging (bottom) of ICG (250 $\mu\text{g/mL}$) and folic-AIEgen (40 $\mu\text{g/mL}$) in PBS at 37 $^{\circ}\text{C}$. (D) Quantitative analysis of the fluorescence signals of ICG and folic-AIEgen in fluorescence imaging. (E) Time-dependent fluorescence imaging of ICG (NIR, 808 nm, 250 $\mu\text{g/mL}$) and folic-AIEgen (365 nm, 40 $\mu\text{g/mL}$) subjected to continuous laser irradiation. (F) Quantitative analysis of the fluorescence signals of ICG and folic-AIEgen in fluorescence imaging.

During a surgical resection of SLN, the invisible NIR fluorescence signals of ICG-stained SLNs could only be detected by a NIR fluorescence imaging system, which consists of NIR excitation light, collection optics, filtration, sensitive CCD camera and specialized software for quantitative analysis of the results, and finally displayed on surgeon's display monitor

(Supplementary Fig. 15). For folic-AIEgen-administered group, folic-AIEgen-stained SLNs were visible to naked eyes only under the illumination of a portable UV lamp, without any supports from external imaging equipment. Notably, the fluorescence signals of folic-AIEgen-stained SLNs via fluorescence imaging system were highly consistent with the visual localization. Compared with the fluorescence/NIR imaging-guided operation requiring expensive equipment and highly-training surgeons, this folic-AIEgen based strategy could provide a more feasible and efficient operation for IGS. Thus, we believe this work will help the popularization of IGS worldwide, especially in developing countries where precision imaging-guiding surgery is urgently needed.

Supplementary Figure 15. Schematic illustration of ICG and folic-AIEgen guided SLN dissection in nude mice. NIR fluorescence imaging of ICG (250 $\mu\text{g}/\text{mL}$, 25 μL) administered mice with skin removal was imaged by NIR fluorescence imaging system. Photographs and fluorescence imaging of folic-AIEgen (40 $\mu\text{g}/\text{mL}$, 25 μL) administered mice with skin removal were recorded under UV light and IVIS imaging system, respectively.

In terms of the damage, UV 385 nm is located in UVA that is the least harmful UV wavelength. Notably, 25 mins cumulative irradiation of 365 nm UV light (3.5 mW/cm^2 intensity) did not cause potential influence on the human cell in terms of gene expression [3]. The UV light source (365 nm UV LED, PKG LEUVA35T01RL00) used in our study has a UV

intensity of 10 mW/cm² with a distance of 25 cm. Therefore, the dose applied in this study, 10 mW/cm² for 3 mins may only induce mild influence on the surgical area. To well demonstrate this, we further evaluated the safety of surgical procedures and the dose of UV irradiation used during the surgery in a mouse model. Mice from the surgery group and surgery + UV illumination group (mice were exposed to 15 minutes of UV light) behaved normally for up to 30 days after administration. Notably, no detectable damage or any inflammatory lesions were found in major tissues (heart, liver, spleen, lung, kidney, intestine, skin and muscle) after UV light exposure in both groups, with no significant difference to the control group (Supplementary Fig. 6). These results further demonstrate the safety of folic-AIEgen based image-guided operation in animal models. The detailed description and results are presented on page 8, line 207-214 and supporting information, Supplementary Fig. 6.

Supplementary Figure 6. Histological images of the major organs from the mice (n = 6) 30 days after different treatments. Representative H&E staining images of heart, liver, spleen, lung, kidney, intestine, skin and muscle. Scale bar = 50 µm.

Reference

- [1] Enggalhardjo, M., Wahid, S., Sajuthi, D., Yusuf, I. Effect of adipose-derived mesenchymal stem cells in photoaging Balb/C mouse model. *AM. J. Med. Bio. Res.* **3**, 48-52 (2015).
- [2] Meinhardt, M., Krebs, R., Anders, A., Heinrich, U. & Tronnier, H. Wavelength-dependent penetration depths of ultraviolet radiation in human skin. *J. Biomed. Opt.* **13**, 044030 (2008).

[3] Wong, D. Y., Ranganath, T. & Kasko, A. M. Low-dose, long-wave UV light does not affect gene expression of human mesenchymal stem cells. *PLoS one* **10**, e0139307 (2015).

Comment 2: *There are also major issues related to folate targeting experiments. First, in vitro experiments show that free folate can block the cell targeting by the AIE NPs. However, in the caption of Figure S4, no information about concentrations of the used free folate and the NPs is provided. In the Methods section, I could find that the used concentration is 1 mM, which is 10,000,000-fold (!) higher compared to the affinity of the receptor to the folate mentioned on page 6 (0.1 nM). At this high concentration, folate can block any sort of binding to the cells surface, so it does not really prove the specificity to the receptor. The second issue is the lack of experimental data in vivo on control NPs without folate ligand. It is hard to evaluate the importance of specific targeting without this control. It should be added at least for the key experiments with xenograft tumors.*

Response: Thanks for reviewer's defined question. As mentioned in the main text, the affinity of folate to FR has been widely accepted and confirmed. In the current work, **Supplementary Fig. 4** well demonstrates the specific-binding of folic-AIEgen to FR-positive SKOV-3 instead of HaCaT negative cell line. Thus, the addition of excess folic acid is aiming to confirm the FR-targeting further. Although the concentration of folate (1 mM) is high, it cannot entirely prevent the FR-binding nanoparticles' internalization, with 40~50% decreases. Notably, more than 90% of folic-AIEgen was blocked at the same condition, showing a better FR-specificity binding [1, 2]. Moreover, in many reported free folate competition studies, 1 mM of folate was applied as a general condition during the incubation [3-5].

We further studied the tumor-targeting ability of folic-AIEgen to intraperitoneal tumors in the SKOV3 xenograft mouse model. The fluorescence signal of folic-AIEgen accumulated in mice's peritoneal tumors was significantly stronger than that of AIEgen without folic acid modification (**Supplementary Fig. 8**). Furthermore, the ex vivo fluorescence images of intestinal tissues with tumors showed that the fluorescence signal of folic-AIEgen treatment group was strong, mainly distributed in tumors. In contrast, the signal of AIEgen treatment

group was much weaker and randomly distributed in either tumor or adjacent normal tissues. The above illustrates the specific targeting ability of folic-AIEgen for SKOV3 xenograft tumors. The detailed description and results are presented on page 11, line 288-299 and supporting information, Supplementary Fig. 8.

Supplementary Figure 8. Specific tumor targeting ability of folic-AIEgen in the intraperitoneal SKOV3 xenograft mouse model (n = 6). (A) In vivo fluorescence imaging of mice (top) 24 hours after intraperitoneal injection of AIEgen or folic-AIEgen (100 μL, 40 μg/mL) and Ex vivo fluorescence imaging of the intestine tissues with tumors collected from mice (bottom). (B) Quantitative analysis of the fluorescence signals of the in vivo and ex vivo images. ** presents p<0.01.

Reference

- [1] Jin, H. *et al.* Folate-chitosan nanoparticles loaded with ursolic acid confer anti-breast cancer activities in vitro and in vivo. *Sci. Rep.* **6**, 1-11 (2016).
- [2] Ren, D., Kratz, F. & Wang, S.-W. Engineered drug-protein nanoparticle complexes for folate receptor targeting. *Biochem. Eng. J* **89**, 33-41 (2014).
- [3] Lee, R. J., Low, P. S. Folate-mediated tumor cell targeting of liposome-entrapped doxorubicin in vitro. *Biochim. Biophys. Acta* **2**, 134-144 (1995)
- [4] Yamada, A., Taniguchi, Y., Kawano, K., *et al.* Design of folate-linked liposomal doxorubicin to its antitumor effect in mice. *Clin. Cancer Res.* **14**, 8161-8168 (2008).

[5] Wu, X., Wang, Z., Hu, B., Ran, H., Li, P., & Xu, C., *et al.* Targeting an ultrasound contrast agent to folate receptors on ovarian cancer cells. *J. Ultras. Med.* **29**, 609-614 (2010).

Comment 3: Scheme 1 is not sufficiently informative and specific. It should be improved and converted into a Figure by including the chemical structure of the dye used, presentation of the nanoparticle and its basic optical characteristics. For the moment, this key information is present in SI, but it would be much better for the reader if this information is presented in a figure of the main text.

Response: Thanks for the reviewer's suggestion. We have revised Scheme 1 to include this vital information for a better understanding.

Scheme 1. From bench to bedside of AIEgens. Schematic illustration of AIEgens for SLNs biopsy and detection of tiny tumors, from murine, rabbit, to nonhuman primate models, and their potential value for clinical translation.

Comment 4: *Important technical information is missing in multiple figure captions: concentration of components used (Figures 1-6) and excitation wavelength used in microscopy experiments (Figures 1-4). Similar problem is with figure captions in the supporting information.*

Response: Thanks for pointing this to us. We have revised these captions, and details of the experiment have been added in the revised manuscript.

Comment 5: *Page 6: the size by DLS of 20.3 nm is too precise. Moreover, the error should be provided for the size values presented on this page.*

Response: Thanks for reviewer's comment. A diameter of NPs, 20.3 ± 1.9 nm was provided, with a PDI < 0.16. These could be found on page 5, line 140.

Comment 6: *Page 7: "A small volume of samples was administrated into the front footpad of nude mice and allowed to move freely." For a broad readership it will not be clear whether the injection was done into a blood circulation or a lymphatic system. Moreover, value of the volume and the total dose of the material injected should be provided here.*

Response: Thanks for reviewer's question. More details have been added for a better understanding, which could be found at page 6, line 167-169, the sentence "Folic-AIEgen (25 \$\mu\$ L, 40 \$\mu\$ g/mL) was subcutaneously injected into the front footpad of nude mice for real-time visualization of lymphatic drainage patterns."

Comment 7: *Methods section (page 21): the protocol for the formulation of folate modified NPs (folic-AIEgen) should be provided.*

Response: Thanks for the reminder in method section. A revision has been made by adding the synthesis method of current NPs, which could be found on page 15, line 417-422.

Reviewer #2:

The authors ignore the fundamental founding body of literature for fluorescence guided surgery (FGS) which significantly undercuts the novelty of the present manuscript. Please see Reviewer Attachment for an example of this literature. In addition the authors do FGS experiments with luciferase photon counting for FGS, which has no relationship to the clinic since photon counting for FGS in the clinic is not practical.

Response: Thanks for reviewer's comment. Firstly, the luciferase photon counting used in this study aim to demonstrate the co-localization of tumor bioluminescence with folic-AIEgen fluorescence, indirectly verifying the sensitivity and accuracy of image-guided surgery based on folic-AIEgen for the detection of tiny tumors. Just because photon counting is not practical in clinical practice, it further highlights the prospect of folic-AIEgen in clinical translations.

In addition, the systematic comparison between folic-AIEgen and potential-clinical probes has been supplemented and presented in our study (Supplementary Table 3&4, Fig 14-16), with emphasis on the advantages of folic-AIEgen over ICG, one of the most frequently employed near-infrared (NIR) fluorophores for FGS.

Properties	ICG, NIR fluorescence ~850 nm	AIEgens, visible fluorescence ~540 nm
Component	Small organic dyes (<1 KD)	Fluorescent nanoparticles (Diameter: ca 20 nm)
Dispersibility	Dispersed in sterile water but easily aggregated in physiological saline	Well dispersed in both sterile water and physiological saline
Fluorescent stability	 1. Decrease to <40% of initial fluorescence intensity within 24 h at 37 °C; 2. Rapid photobleaching to ~52% in 105 min under continuous laser irradiation 	 1. Remain stable during 3-day storage; 2. Possess superior photostability, preserving ~80% during 105 min continuous laser irradiation
Real-time surgical operation	 1. NIR fluorescence imaging system, including NIR excitation light, collection optics, filtration, NIR camera, color camera; 2. Assisted by surgeon's guidance due to undetectable NIR fluorescence seen by naked eyes 	Portable UV lamp
Medical equipment cost	High cost (~100,000 US dollars)	Low cost (100-500 US dollars)
Medical training	Long-term training time	No additional training Simple operation procedure

Supplementary Table 3. Systematically Comparison of ICG dye and AIEgens.

Typical example	Method	Diameter	Advanced performances	Limitations
^{99m} TcTechnetium nanocolloids	Nuclear imaging	50~3000 nm	Detect deep-seated targets (~25 mm)	 1. Large sizes; 2. Slow clearance from injection site; 3. Slow physiologic transport within lymphatics
ICG	MR fluorescence imaging	<1 KD	Detect superficially located targets (~5 mm)	Extravasation and nonspecific tissue scattering
Evans blue	Visible blue color	<1 KD	Aid visual identification under white-light illuminations	 1. Limited tissue penetration; 2. Lose visibility when intraoperative bleeding; 3. Extravasation and nonspecific tissue scattering
AIEgens	Naked eye visible fluorescence	~20 nm	 1. Aid visual identification under UV-light illuminations; 2. Visualize the draining lymphatics in real time 	Limited tissue penetration

Supplementary Table 4. Advanced performances and limitations of AIEgens and typical contrast agents.

ICG, as the most common dye could offer a desirable signal-to-background ratio and tissue-penetration depth, which has been widely employed in clinical applications. Nevertheless, the ICG-based imaging is also subjected to the finite quantum yield, short-term retention in tumor and nonspecific interaction with cells. Besides, the particular camera and systematic training required for NIR probes usage further restrain the promotion of ICG. Therefore, novel intraoperative imaging modalities with tumor-specific targeting are in urgent need.

In our study, the as-prepared folic-AIEgen could actively target folate receptor type alpha (FR α) overexpressing SKOV3 and Hela cancer cells and tissues with mild cytotoxicity. Furthermore, folic-AIEgen could successfully support SLNs biopsy (from mice to rhesus macaque) and image-guided surgery for tiny SKOV3 and Hela tumors after peritoneal dissemination. Our findings indicate that folic-AIEgen could serve as an effective and convenient fluorescent probe for SLNs biopsy and precisely detection of tiny tumors.

Notably, it is the first time that AIEgens has been successfully applied in the rhesus macaque model. This study has provided valuable data for preclinical studies in non-human primate, which would greatly facilitate further progress of folic-AIEgen and other AIE-based luminescence probes in clinical translations.

Reviewer #3:

This manuscript describes research utilizing a novel folic-AIEgen nanoparticles for the intraoperative visualization of lymph vessel, draining lymph nodes, as well as induced carcinomas and their metastases in animal models. The technique was demonstrated in a rhesus macaque by injecting the AIE luminogen into mammary tissue and then following the draining lymph vessels to the illuminated axillary lymph node which was then excised. This technology has great translational value for use in human therapeutic surgical interventions. The authors are correct in noting the surgical time reduction, surgical efficacy, and

identification of small metastases may be improved with targeted fluorescent image guidance. All of these improve patient outcomes by reducing surgery time and reducing post-operative metastatic recurrence. I was asked to focus my attention on the rhesus macaque lymph node excision section, although several of my comments apply to the entire document.

Comment 1: *More information or references are needed about the relative safety of AIE luminogen when administered into tissues. Specifically, how the agent is metabolized and/or eliminated and how this compares to ICG metabolism and elimination. This is important for patient safety.*

Response: Thank you for your crucial suggestions to improve the quality of our manuscript. Some revisions have been made to highlight the safety of AIEgen for in vivo application, for instance, at page 4, line 102-107. Moreover, additional studies were conducted to compare the metabolism of folic-AIEgen with that of ICG. The detailed description and results are presented on page 13, line 365-369 and Supplementary Fig. 16.

As shown in Supplementary Fig. 16, as a small-molecule dye, ICG was rapidly metabolized. The short-term tissue retention of ICG was only observed in the liver and kidneys, presumably because one of the possible ICG membrane carrier molecules, bilitranslocase, is only expressed in these two organs [1, 2]. Compared with ICG, folic-AIEgen was mainly excreted through the liver and stayed in tissues for a longer time. However, it was metabolized entirely at 96 hours post-injection, further proving the biosafety of folic-AIEgen in vivo.

Supplementary Figure 16. Biodistribution of ICG and folic-AIEgen in nude mice after intravenous administration. Ex vivo fluorescence imaging of major tissues (brain, heart, lung, liver, kidney, spleen, bone and muscle) of mice (n = 6) collected at different time points after intravenous injection with ICG (250 $\mu\text{g/ml}$, 100 μL) or folic-AIEgen (40 $\mu\text{g/ml}$, 100 μL). Images in different groups were required under the corresponding instrumental conditions (Ex: 745 nm/Em: ICG for ICG group, Ex: 430 nm/Em: GFP for folic-AIEgen group).

Reference

- [1] Sottocasa, G. L., Passamonti, S., Battiston, L., Pascolo, L., Tiribelli, C. Molecular aspects of organic anion uptake in liver. *J. Hepatol.* **24**, 36-41 (1996).
- [2] Elias, M. M., Lunazzi, G. C., Passamonti, S., Gazzin, B., Miccio, M., Stanta, G., Sottocasa, G. L., Tiribelli, C. Bilirubin localization and function in basolateral plasma membrane of renal proximal tubule in rat. *Am. J. Physiol.* **259**, 559-564 (1990).

Comment 2: *The entire manuscript needs to be edited by a native speaker of English as there are atypical word choices and expressions throughout.*

Response: Thanks for the reminder in English writing. A series of revisions have been made throughout the main tests.

Comment 3: *Line 491: recommend changing title to be more specific, "Fluorescence-guided excision of a mammary SLN in rhesus macaque"*

Response: Thanks for the valuable suggestion. We have revised this section accordingly base on your comments.

Comment 4: Line 492-512: I've rewritten this section and suggest it read as follows. I made some assumptions about the surgical technique and monitoring parameters, so please check for accuracy. Questions and suggestions are provided in parentheses.

The animal subject was determined to be normal and active during the pre-operative assessment conducted the day prior to surgery. Food was withheld for 8 hours prior to surgery and water was withheld for 2 hours prior to surgery. General anesthesia was induced with ketamine HCl (10 mg/Kg, IM) with atropine (0.03 mg.Kg, IM). Once anesthetized, the animal was endotracheally intubated and anesthesia was maintained with 1-2% isoflurane mixed with 100% oxygen. Heart rate, body temperature, oxygen saturation (SpO₂), and end-tidal CO₂ (ETCO₂) (authors - I think you were monitoring ETCO₂, which is an approximation of PaCO₂. If you were monitoring PaCO₂, please specify the blood gas machine used) were monitored by a physiological monitor (iPM 12 Vet, Mindray, China). The hair from the right axilla was clipped and the axillary region was sterilely prepped for surgery using (authors - specify agent: betadyne? or chloraprep?? - and describe sterile prep here). Next, a 5cm incision (??) was made in the right axilla to visualize the region of the fat pad and lymph nodes. Folic-AIEgen was then injected into the mammary tissue of the right areola. Using a 365 nm UV light was directed to the incision site and the progression of the fluorescent indicator through the lymph vessels and into the draining lymph node (or SLN) was recorded with a digital camera. The once illuminated and identified by the surgeon, the SLN was excised. The skin incision was then sutured (add suture material, gauge, pattern) and a bandage was applied. Maintenance anesthesia was discontinued, the animal was recovered from anesthesia, and returned to the home cage. A prophylactic antibiotic (cefazolin, 25 mg/Kg IM) was administered at the time of surgery and every 24 hours for 7 days post-op (authors - please check this as cefazolin is normally administered every 8-12

hours rather than every 24 hours). Analgesia was provided with buprenorphine administered pre-op and then every 8 hours for 3 days post-op (authors - please check the dose administered here - the published dose range is 0.01-0.03 mg/Kg every 8-12 hours, which is about 4 times more than you are reporting here. The dose reported would provide little, if any, analgesia). Post-operative animal behavior, appetite, body weight, and blood work were monitored and reported as normal. (reference Fig. S7 here)

Response: We feel great thanks for your professional suggestion. Your revision could improve the quality of our paper remarkably. We have revised this section accordingly based on your comments, and added the details of experiment.

Reviewer #4:

This submission by Zhong et al. reported aggregation-induced emission luminogens for image-guided surgery in multiple animal models including mouse, rabbit and rhesus macaque through targeting folate receptor. The as-prepared folic-AIEgen was characterized with appropriate physicochemical properties and biocompatibility. In vivo toxicity was also determined. The authors first performed image-guided biopsy in mice, rabbits and rhesus macaque which showed visualization of SLNs. Then the authors tested this strategy in multiple tumor models and demonstrated the efficiency of dissecting the tumors and metastatic SLNs, especially in the survival studies. Overall, this is an interesting study based on solid development an image-guided strategy to help the surgical operation. However, there are major concerns associated with the current version of submission which are detailed below. Taken together, this is an interesting study and the results have the potential for future translation. Given the major issues discussed above, I would recommend a major revision prior to the acceptance for a publication on Nature Communications.

Comment 1: *The authors proposed to use folate receptor as a targeting strategy, which is a widely used target for multiple cancer types. However, different subtypes of folate receptor could have different applications in certain tumors. It is necessary to be more specific on this. Moreover, the levels of folate receptor were not characterized in all the tumors which is necessary for this study.*

Response: Thanks for the reviewer's constructive comment. As you mentioned, the folate receptor has been widely applied for tumor targeting due to its overexpression on cancerous cells. Both SKOV3 and Hela cancer cell lines used in our study to construct tumor models overexpress folate receptor (FR) type alpha (FR α). We have specialized this section accordingly based on your comments, which could be found at page 5, line 122-123. In current studies, the application of folic-AIEgens among different xenograft models is aiming to evaluate its feasibility and applicability in general IGS instead of comparison among cells with different expression levels of folate receptor. For this purpose, previous studies on FR expression are able to support our current study. Additionally, we believe it is vital to further investigate and improve our IGS contrast agent in various FR subtypes. As our future works, the specific targeting for FR subtypes will be explored.

Reference

- [1] Toffoli, G., Cernigoi, C., Russo, A., Gallo, A., Bagnoli, M., Boiocchi, M. Overexpression of folate binding protein in ovarian cancers. *Int. J. Cancer* **74**, 193–198 (1997).
- [2] Lu, Y., Low, P. S. Immunotherapy of folate receptor-expressing tumors: review of recent advances and future prospects. *J. Control Release* **91**, 17-29 (2003).

Comment 2: *One major concern is the lack of demonstration of targeting specificity in this study. There was no control AIEgen tested in the same animal models to validate folate receptor mediated uptake.*

Response: We feel great thanks for your professional suggestion. An additional study on the tumor-targeting ability of control AIEgen and folic-AIEgen in the SKOV3 xenograft mouse

model has been finished and provided. The detailed description and results are presented on page11, line 288-299 and supporting information, Supplementary Fig. 8.

The folic-AIEgen accumulated in mice's peritoneal tumors presented a stronger fluorescence signal than that of AIEgen without folic acid modification (Supplementary Fig. 8). Furthermore, the ex vivo fluorescence images of the intestinal tissues with tumors showed that the folic-AIEgen treatment group's fluorescence signal was strong, mainly distributed in the tumors. In contrast, the AIEgen treatment group's signal was much weaker and randomly distributed in either tumor or adjacent normal tissues. The above illustrates the specific targeting ability of folic-AIEgen for SKOV3 xenograft tumors.

Supplementary Figure 8. Specific tumor targeting ability of folic-AIEgen in the intraperitoneal SKOV3 xenograft mouse model (n = 6). (A) In vivo fluorescence imaging of mice (top) 24 hours after intraperitoneal injection of AIEgen or folic-AIEgen (100 μ L, 40 μ g/mL) and Ex vivo fluorescence imaging of the intestine tissues with tumors collected from mice (bottom). (B) Quantitative analysis of the fluorescence signals of the in vivo and ex vivo images. ** presents $p < 0.01$.

Comment 3: Folate receptors such as *FR β* are also overexpressed by myeloid cells and macrophages, which may complicate the operation if using the proposed strategy. Can the authors comment on this?

Response: Thanks for the reviewer's question. Compared with the FR α expressing on tumor cells, FR β does express on myeloid cells and macrophages. Notably, the most FR β -overexpressing macrophages are tissue resistant macrophage and tumor-associated macrophage (TAM), according to recent studies [1, 2]. Although the TAM may attract some signal from FR-targeting AIEgens, the TAM is located in the same position as cancer cells, which will not affect IGS outcomes. In consideration of the convenience and efficiency during surgery, a local administration was generally recruited in the current IGS system after a pre-diagnosis. In such situation, the AIE intensity cannot be influenced by the nonspecific binding on tissue-resistant macrophages due to the limited circulation

Reference

- [1] Samaniego, R. *et al.* Folate Receptor β (FR β) Expression in Tissue-Resident and Tumor-Associated Macrophages Associates with and Depends on the Expression of PU. 1. *Cells* **9**, 1445 (2020).
- [2] Tie, Y. *et al.* Targeting folate receptor β positive tumor-associated macrophages in lung cancer with a folate-modified liposomal complex. *Signal Transduct. Tar.* **5**, 1-15 (2020).

Comment 4: *In all the studies, the authors had very few replicates with n=3. It is necessary to increase the rigor of the submission with more replicates.*

Response: Thanks for the reviewer's comment. Due to the limited number of large animals, especially rhesus macaque, we can only test our AIEgen system in a certain number of animals. As for the small animal, the "n=3" stands for triplicate studies instead of mice number. All data are presented as means and standard deviations of at least 3 independent experiments. Generally, there were 3-5 mice for each study. In other words, there were about 9-15 mice in total for each group. To avoid the misunderstanding, we have made some modifications accordingly.

Comment 5: *The AIEgen used in this study was approximately 20 nm. There could be significant liver or spleen accumulation. However, it was not observed in Figure S10. It will help the audience to better understand the potential of as-developed strategy if the authors include the biodistribution data following intravenous injection.*

Response: Thank you for your vital suggestions to improve the quality of our manuscript. An additional study on the biodistribution and metabolic analysis of folic-AIEgen after intravenous administration has been finished and provided. The detailed description and results are presented on page 13, line 365-369 and Supplementary Fig. 16.

Significant hepatic aggregation was observed after intravenous administration of folic-AIEgen, and the signal gradually decreased over time. These results indicated that folic-AIEgen was mainly excreted through the liver and completely metabolized at 96 hours post-injection, further proving the biosafety of folic-AIEgen in vivo.

Supplementary Figure 16. Biodistribution of folic-AIEgen in nude mice after intravenous administration. Ex vivo fluorescence imaging of major tissues (brain, heart, lung, liver, kidney, spleen, bone and muscle) of mice (n = 6) collected at different time points after intravenous injection with folic-AIEgen (40 $\mu\text{g}/\text{ml}$, 100 μL). Images were required under the same instrumental conditions (Ex: 430 nm/Em: GFP).

Comment 6: *Local administration was mostly used in the manuscript. However, it would be difficult to do it in clinics. Can the authors comment on this aspect, especially the detection of distant metastasis?*

Response: Thanks for the reviewer's constructive comment. Indeed, folic-AIEgen fails to act as a contrast agent for primary diagnosis due to its UV-based imaging approach. However, this feasible and efficient imaging system shows a great advantage in IGS, which is based on pre-surgical imaging diagnosis (such as PET/CT or MRI). After an imaging location, IGS via local administration is manageable even in clinics. More importantly, folic-AIEgens was also employed for imaging and help a precision surgery via i.v. injection. In this study, distant metastasis (lesions with size $1 \leq \text{mm}$; $2 \leq \text{mm}^3$) could be successfully dissected via IGS, as shown in **Figures 5&6**. Depending on the cancer type, folic-AIEgens could be used via local and systemic administration, both of which are able to meet the requirement of precision IGS.

Comment 7: *There are some typos in the manuscript. For example, in scheme 1, it should be "bedside", not "beside".*

Response: Thanks for pointing this to us. We have made corrections in the revised version.

REVIEWERS' COMMENTS

Reviewer #1 (Remarks to the Author):

In the revised manuscript, the authors addressed well all my concerns and made sufficient number of new experiments to support their claims. Now I can recommend this manuscript for publication in Nature Communications in the present form.

Reviewer #3 (Remarks to the Author):

This is a review of the authors' revisions of the manuscript. My concerns were addressed (Reviewer #3) adequately. However, the atypical word choices and descriptions that I described in my original remarks (weaknesses, item #2) will require more editing prior to publishing.

Reviewer #4 (Remarks to the Author):

The authors have addressed all the comments.

We sincerely thank the editor and all reviewers for their valuable feedback that we have used to improve the quality of our manuscript entitled “Aggregation-Induced Emission Luminogens for Image-Guided Surgery in Non-Human Primates” (Manuscript No. NCOMMS-20-41343B). We have studied comments carefully and have made corrections which we hope meet with approval. Our response is given in normal font and changes/additions to the manuscript are given in the red text.

Reviewer #1 (Remarks to the Author):

In the revised manuscript, the authors addressed well all my concerns and made sufficient number of new experiments to support their claims. Now I can recommend this manuscript for publication in Nature Communications in the present form.

Response: We appreciate Reviewer #1’s positive comments.

Reviewer #3 (Remarks to the Author):

This is a review of the authors' revisions of the manuscript. My concerns were addressed (Reviewer #3) adequately. However, the atypical word choices and descriptions that I described in my original remarks (weaknesses, item #2) will require more editing prior to publishing.

Response: We appreciate Reviewer 3’s positive comments. Thanks for pointing this to us. We have made corrections for the word choice and description and highlighted them as red in the revised version.

Reviewer #4 (Remarks to the Author):

The authors have addressed all the comments.

Response: We appreciate Reviewer #4’s positive comments.